# ELMO2 is an essential regulator of carotid artery development

Athira Suresh [1], Kai Kruse [2], Hendrik Arf [1], Rodrigo Diéguez-Hurtado [1] & Ralf H. Adams [1]

Engulfment and cell motility 2 (ELMO2) is a cytoskeletal adaptor protein necessary for cell migration and apoptotic cell removal. Loss-of-function mutations in *ELMO2* cause intraosseous vascular malformation (VMOS), a human disease involving progressive expansion of craniofacial bones in combination with anomalies in blood vessels that emerge from the external carotid artery, as well as aneurysms in the internal carotid artery. Here we show that global inactivation of *Elmo2* in mice leads to midgestation embryonic lethality due to dilation of the 3rd pharyngeal arch arteries and aneurysm formation in the common carotids. These vascular malformations are associated to defects in vascular smooth muscle cell organization and are phenocopied upon neural crest-specific deletion. In vitro experiments further confirm that ELMO2 regulates vascular smooth muscle cell adhesion, spreading and contractility through Rac1 activation and modulation of actin dynamics. Our findings provide new insights into the biological function of ELMO2 with relevant implications for understanding VMOS pathogenesis and raise the possibility of vessel-targeted diagnostic and treatment strategies.

Defective regulation of blood vessel growth and patterning is the cause of numerous hereditary but also spontaneous human diseases, and recent insight into the underlying molecular defects has enabled targeted therapeutic approaches[1]. Intraosseous vascular malformation (VMOS) is a very rare and poorly understood autosomal-recessive disorder involving progressive overgrowth of the mandible, maxilla, and other craniofacial bones. Defects also involve vascular alterations inside the affected skeletal elements[2,3] and in the carotid artery or its side branches, which are found enlarged or affected by aneurysms[4]. The causal relationship between vascular and skeletal malformations in VMOS remains unknown, but it has been suggested that dilated blood vessels inside lesions are insufficiently covered by vascular smooth muscle cells (VSMCs)[3,5]. Similar VSMC defects are associated with dilation and aneurysm formation in the dorsal aorta[6].

The identification of loss-of-function mutations in the human gene *ELMO2* (Engulfment and cell motility 2) as a cause of VMOS was an important milestone for understanding the disease[2,7,8]. ELMO2 is a cytoplasmic protein known to regulate cytoskeletal dynamics by Rho-Rac regulation[9]. It belongs to the ELMO domain containing (ELMOD) family[10] and is conserved across different species[9,10]. Mammalian ELMO proteins (ELMO1, 2, and 3) share multiple homology domains such as Ras binding domain (RBD), Armadillo repeats, ELMO domain, pleckstrin homology (PH) domain, and C-terminal proline-rich repeats[9,10], which allow binding to Rho[2,9–12], integrin-linked kinase (ILK)[13–15], dedicator of cytokinesis (DOCK)[2,9–12,16,17], and spectraplakin[18–20]. These interactions link ELMO proteins to Rac1 signaling and the control of cytoskeletal dynamics during cell adhesion, junction formation, establishment of cell polarity, and cell migration[9,12–16,21,22].

While all ELMO proteins show high sequence homology and partially redundant functions[2,10,22–24], global deletion of *Elmo1* in mice only affects Sertoli cells in the male reproductive system[11], whereas *Elmo2* knockout mice die during embryonic gestation due to a yet uncharacterized phenotype[17]. The roles of *Elmo2* in the vascular and skeletal system remain unexplored although it is known that signaling defects

[1]Max Planck Institute for Molecular Biomedicine, Department of Tissue Morphogenesis, Münster, Germany. [2]Max Planck Institute for Molecular Biomedicine, Bioinformatics Service Unit, Münster, Germany. ✉e-mail: rodrigo.hurtado@mpi-muenster.mpg.de; ralf.adams@mpi-muenster.mpg.de

in functionally-associated proteins such as integrins, Ilk and Rac1 are responsible for cardiovascular and craniofacial abnormalities[25–30].

Vascular malformations are non-neoplastic congenital anomalies affecting blood vessels. They result from errors in the development program of the vascular system due to somatic or germline prenatal mutations[31]. Most sporadic vascular malformations are caused by somatic mutations that activate the RAS/MAPK/ERK and/or the PI3K/AKT/mTOR signaling pathways. Familial malformations are caused by loss-of-function mutations in genes related to TGFβ signaling, RASA1, glomulin, or CCMs (cerebral cavernous malformations)[32]. Interestingly, vascular anomalies frequently occur in the head and neck[33], and usually involve defects in the control of endothelial or mural cells, which together form the vascular wall.

The vasculature of the head and neck region derives mostly from the pharyngeal arch arteries (PAAs), which are a series of six paired arteries that sequentially emerge from embryonic day (E) 9.5 to E10.0 in mice and connect the aortic sac to the paired dorsal aortae[34]. During development, the PAAs undergo complex remodeling to form the major arteries supplying the head, neck, and upper thorax. The endothelial cells (ECs) of the PAAs are derived from *Mesp1*-lineage positive mesoderm[26], while the surrounding mural cells, namely pericytes and vascular smooth muscle cells (VSMCs), arise from the neural crest[35]. In addition, neural crest cells (NCCs) also give rise to most connective tissues in the head and neck region, including bone and cartilage.

Here, we have used mouse genetics, advanced confocal and light-sheet microscopy, single-cell RNA-sequencing (scRNA-seq), in vitro cell culture, and biochemistry approaches to systematically investigate the functional role of Elmo2 during mouse embryonic development. We show that the gene product is indispensable for vascular morphogenesis of the 3rd PAA through the control of the contractile properties of neural crest-derived VSMCs and highlight that vascular defects may be a primary cause of the lesions observed in VMOS patients.

## Results

### Loss of *Elmo2* leads to carotid artery aneurysm and embryonic lethality

In order to understand the in vivo function of *Elmo2*, global knockout mice (*Elmo2⁻/⁻*), obtained after ubiquitous Cre-mediated recombination of a "knockout-first" allele (Supplementary Fig. 1a, b), were compared to control littermates (*Elmo2⁺/⁺*) at different embryonic stages. Efficient deletion of *Elmo2*-encoded transcripts and protein were confirmed by RT-qPCR and Western blot, respectively (Supplementary Fig. 1c, d). The first macroscopic evidence of deleterious phenotypic alterations (Fig. 1a) was detected in E12.5 *Elmo2⁻/⁻* embryos, which show small vascular lesions in the head and neck region. These lesions worsened during the following days of development, giving rise to subcutaneous edema and severe hemorrhages in the cervical region by E13.5 and E14.5. No surviving embryos were obtained beyond this point.

Taking advantage of β-galactosidase expression from the targeted allele, heterozygous (*Elmo2⁺/⁻*) embryos, which develop normally, were assessed by X-Gal staining at E12.5. This approach revealed expression of *Elmo2* in different structures including the dorsal aorta, laryngo-tracheal groove, vagus nerve, sympathetic chain ganglia, trachea, esophagus and pharyngeal arch arteries (Fig. 1b and Supplementary Fig. 1e). In line with this expression pattern, the histological analysis of transverse sections from the cervical region of mutant embryos revealed a severe dilation of the third pharyngeal arch artery (3rd PAA) at E12.5 and of the carotid arteries at E13.5 (Fig. 1c–e and Supplementary Fig. 1f), which in some embryos lead to compression and collapse of the jugular vein as well as presence of blood-filled lymph sacs (Supplementary Fig. 1g).

The complex and dynamic morphogenetic program that shapes hierarchical blood vessel organization in the trunk and cervical area[36] prompted us to analyze the three-dimensional organization of the vascular tree in control and *Elmo2⁻/⁻* embryos from E11.5 to E13.5 using whole-mount staining and light-sheet microscopy. This analysis showed that the first vascular defect in global knockout embryos, namely the dilation of the 3rd PAA, arises at E12.5. This vessel further remodels giving rise to the common carotid arteries[37], which are severely dilated resulting in fusiform aneurysm formation in E13.5 mutants (Fig. 1f). It is worth noting that these defects in vascular shape and diameter are rather specific to the 3rd PAA and only mildly affect other major vessels, including neighboring pharyngeal arch arteries (Supplementary Fig. 2a–e).

### Loss of *Elmo2* leads to alterations in endothelial and vascular smooth muscle cells

To characterize the vascular defects in *Elmo2⁻/⁻* embryos in greater detail, the 3rd PAA and carotid arteries were analyzed by immunofluorescence staining and high-resolution confocal microscopy. This revealed discontinuities in the endothelial lining (Supplementary Fig. 3a) and significant changes in the size and morphology of ECs and their nuclei (Fig. 2a, b) in mutant embryos relative to littermate controls. In addition, defects in the polarized expression of the luminal marker Podocalyxin were observed (Fig. 2c) as well as ectopic expression of the VSMC marker α-smooth muscle actin (αSMA) in ECs (Fig. 2d, e). Despite normal and domain-specific expression of markers for lymphatic vessels (Prox1) and arteries (SOX17) (Figs. 1e and 2a), clusters of ECs in the dilated carotid arteries of *Elmo2⁻/⁻* embryos express detectable levels of Endomucin (Fig. 2f, g), a marker that at early embryonic stages is expressed in the dorsal aorta[38] but quickly becomes restricted to veins and is normally absent from the arterial endothelium.

VSMC contractility is an important regulator of vascular tone in the adult organism, but also in the embryo[39]. We therefore analyzed VSMCs around the 3rd PAA and carotid arteries by immunostaining against proteins associated with the contractile phenotype. This revealed that the expression levels of SM22α, αSMA, and Calponin1, among other markers, are comparable between control and *Elmo2⁻/⁻* embryos (Supplementary Fig. 3b–i), arguing against major defects in VSMC abundance and differentiation. Likewise, no overt differences were detected for Nestin (Supplementary Fig. 3f), whose expression has been associated to the synthetic VSMC phenotype[40]. The staining intensity and distribution pattern of phospho-myosin light chain 2 and 9, functional markers of VSMC contractility, also failed to show obvious changes in the wall of mutant carotid arteries relative to littermate controls (Supplementary Fig. 3g, h).

Although the expression of VSMC identity and differentiation markers appears unaffected, super-resolution confocal microscopy revealed that the normal alignment of αSMA⁺ bundles with respect to the longest axis of the underlying endothelium is severely compromised in *Elmo2⁻/⁻* embryos (Fig. 2h–j). Whereas αSMA⁺ fibers are oriented perpendicular to elongated ECs and thereby to the direction of blood flow in the control E12.5 3rd PAA, both EC elongation and the orientation of αSMA⁺ bundles are disorganized after loss of *Elmo2*. Furthermore, αSMA and SM22α immunosignals, which are strongly concentrated near the subendothelial basement membrane in VSMCs of control carotid arteries, have lost their normal polarization, and multiple peaks of high staining intensity can be detected throughout the *Elmo2⁻/⁻* vessel wall (Fig. 2k, l). The same analysis also confirmed the abnormal expression of VSMC markers in the *Elmo2⁻/⁻* carotid artery endothelium (Fig. 2l).

Next, we assessed whether significant changes in EC or VSMC proliferation are associated with the vessel enlargement in *Elmo2⁻/⁻*

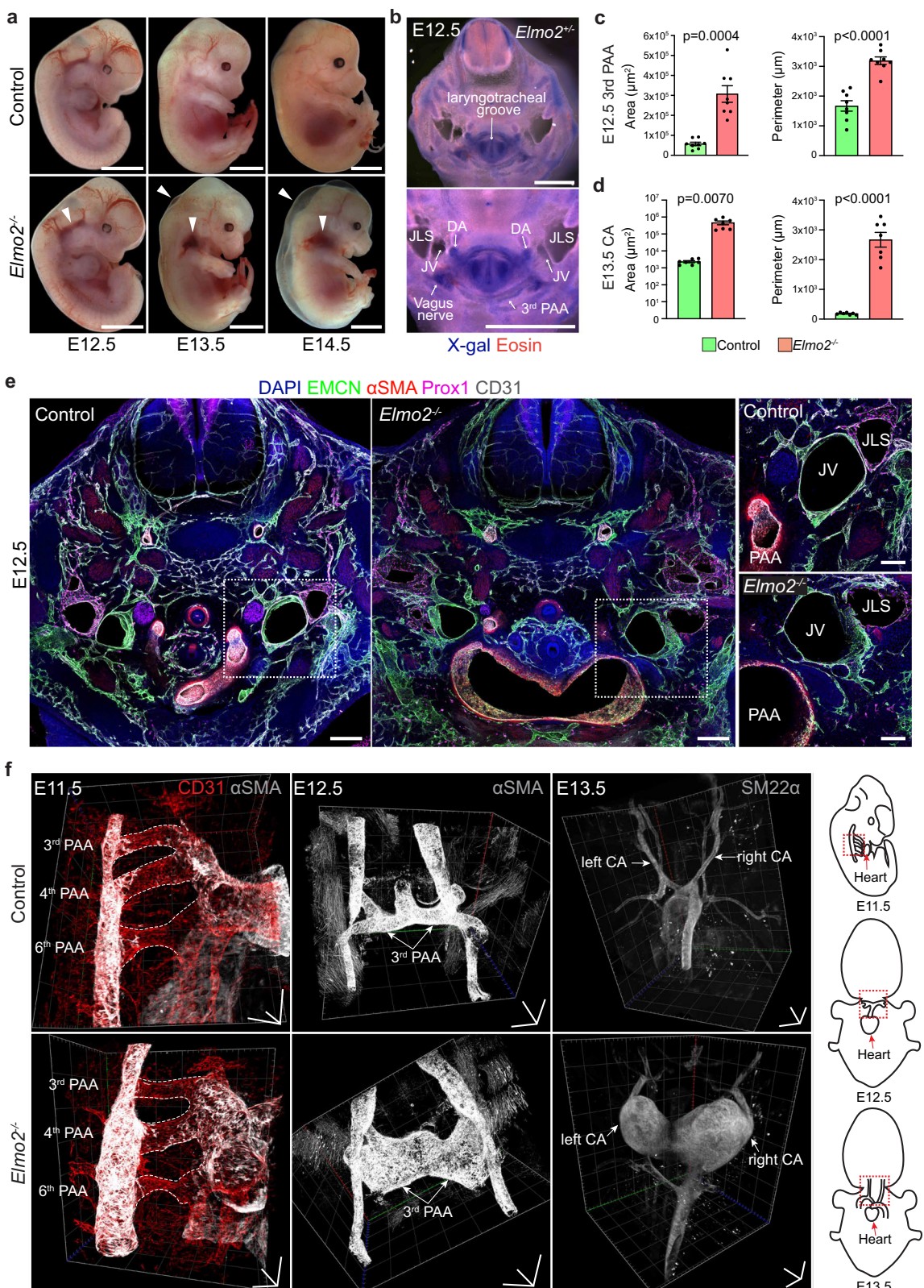

mutants. To this end, we labeled mitotic cells in vivo by injection of 4-Ethynyl-2'-deoxyuridine (EdU) into pregnant females. This approach revealed strong increases in the absolute number of EdU+ ECs and VSMCs in the 3rd PAA of E12.5 mutant embryos (Supplementary Fig. 4a, b). However, the magnitude of these changes is related to vessel enlargement and no longer statistically significant when normalized to vessel perimeter (Supplementary Fig. 4c). In addition, the

number of proliferating ECs and VSMCs in mutants at E11.5 is not altered (Supplementary Fig. 4d–f), suggesting that the increase in proliferation may be a consequence of vessel dilation and disturbed flow rather than the underlying cause of the vascular malformation.

Altogether, these results establish that ELMO2 is required for the normal development of the 3rd PAA and common carotid artery during embryogenesis.

**Fig. 1 | Global deletion of *Elmo2* leads to carotid artery aneurysm and embryonic lethality. a** Representative images of control and *Elmo2⁻/⁻* embryos at different developmental stages showing hemorrhages in the cervical region and subcutaneous edema (white arrowheads) in knockout mice. Scale bars, 2 mm. **b** Cross-section of an E12.5 *Elmo2⁺/⁻* embryo stained with X-gal (blue) and Eosin (red) showing expression of *Elmo2* in the cervical region. Higher magnification (bottom panel) shows relevant vascular structures: dorsal aorta (DA), third pharyngeal arch artery (3ʳᵈ PAA), jugular vein (JV), and jugular lymph sac (JLS). Scale bars, 2 mm. **c** Quantitation of area and perimeter of the 3ʳᵈ PAA in control and *Elmo2⁻/⁻* E12.5 embryos. Mean ± SEM, *n* = 8. Welch's *t*-test (area) and unpaired *t*-test (perimeter). **d** Quantitation of area and perimeter of the carotid artery (CA) in control and *Elmo2⁻/⁻* E13.5 embryos. Mean ± SEM, *n* = 7. Welch's *t*-test. **e** Confocal images of control and *Elmo2⁻/⁻* E12.5 embryos stained for nuclei (DAPI, blue), veins and capillaries (EMCN, green), vascular smooth muscle cells (VSMCs, αSMA, red), lymphatic endothelial cell (EC) nuclei (Prox1, magenta) and ECs (CD31, gray). Higher magnification images correspond to the insets (outlined by white dashed line) in the respective overview images and highlight the 3ʳᵈ pharyngeal arch artery (PAA), jugular vein (JV), and jugular lymph sac (JLS). Scale bars, 200 μm (overview) and 50 μm (higher magnification). **f** Representative light sheet microscopy 3D images of control and *Elmo2⁻/⁻* embryos at different developmental stages stained for ECs (CD31, red) and VSMCs (αSMA or SM22α, gray). Vascular structures of interest are indicated: pharyngeal arch arteries (PAA) or carotid arteries (CA). The schematic on the right shows the orientation of the embryos and the approximate area imaged in the respective developmental stages. Scale bars, 300 μm in each dimension.

## Transcriptomic analysis of *Elmo2* mutants at single cell resolution

To gain insight into the molecular changes resulting from the inactivation of *Elmo2*, we performed scRNA-seq analysis of the 3ʳᵈ PAA and surrounding mesenchyme dissected from control (*Elmo2⁺/⁺*), heterozygous (*Elmo2⁺/⁻*), and mutant homozygous (*Elmo2⁻/⁻*) E12.5 embryos. Integrated analysis of the transcriptome from these samples allowed identification of seven major cell populations with distinct expression signatures and enrichment of specific markers (Fig. 3a and Supplementary Fig. 5a). The most abundant cell type is the mesenchymal stromal cell (MSC) population, which represents more than 70% of all cells analyzed. MSCs are followed by endothelial, immune, and muscle cells, which are found in similar proportions and together represent ~20% of total cells. The remaining cell types are mostly erythrocytes, neurons and epithelial cells (Supplementary Fig. 5b). As expected, *Elmo2* transcript expression is proportionally reduced in heterozygotes and is below the detection threshold in *Elmo2⁻/⁻* samples (Fig. 3b). Among the members of the Elmo family, *Elmo2* has the highest expression, followed by *Elmo1* and *Elmo3* (Supplementary Fig. 5c), which do not show significant compensatory upregulation upon deletion of *Elmo2* (Supplementary Fig. 5d). Furthermore, expression of *Elmo2* is rather homogeneous across the different cell clusters with the highest level found in neurons and the lowest in the erythroid lineage (Fig. 3c).

Considering that the phenotypic changes in *Elmo2⁻/⁻* mutants affect mostly the vascular compartment, the control and homozygous mutant EC populations in our scRNA-seq data were subclustered for deeper analysis. Three main subsets with distinct markers were identified in a two-dimensional (2D) Uniform Manifold Approximation and Projection (UMAP) representation, namely venous, arterial and lymphatic ECs (Fig. 3d and Supplementary Fig. 5e). Interestingly, color-labeling of cells corresponding to the control (*Elmo2⁺/⁺*) or knockout (*Elmo2⁻/⁻*) samples within the subclustered EC dataset highlighted an area characterized by overrepresentation of mutant cells in a specific 2D spatial location within the arterial subcluster (Fig. 3e). In addition, differential gene expression analysis (DEG) allowed the identification of de-regulated genes in *Elmo2⁻/⁻* cells relative to control. Interestingly, when a stringent selection criterion for highly statistically significant values is used (*p*-adjusted < 1⁻¹⁰), only a single gene (*Elmo2*) is downregulated. In contrast, 38 genes are upregulated, 9 of them with a log2 fold change above 2 (Fig. 3f). Notably, all these upregulated genes are either exclusively or primordially expressed in the mutant cell hotspot within the arterial subcluster (Fig. 3g). Among the upregulated genes, *Acta2* and *Tagln* were previously identified during our histological analysis because of their ectopic expression in arterial ECs of mutant embryos (Fig. 2d, e and Supplementary Fig. 3e).

Next, we followed a similar approach for the identification of subpopulations within the mesenchymal stromal cells (MSCs), which are a source of VSMCs during development[41,42]. Both the UMAP representation (Fig. 3h) and marker analysis (Supplementary Fig. 5f) indicate that the differences between the 8 identified MSC subtypes are less defined than those found during the EC subclustering, potentially reflecting ongoing differentiation and incomplete terminal phenotypic specification. Likewise, cellular distribution of control and *Elmo2* mutant cells within the MSC UMAP plot is rather homogeneous (Fig. 3i) and only a few genes were found to be de-regulated (log2 fold change > 2 or < − 2) when a cut-off for highly statistically significant differences (*p*-adjusted < 1⁻¹⁰) is applied (Fig. 3j). Using this criteria, 4 downregulated (*Elmo2*, *Hoxb6*, *Car2* and *Capn11*) and 3 upregulated genes (*Cnmd*, *Matn1* and *Acan*) were identified, without clear functional relationships among them. A similar profile of very limited or non-significant changes in gene expression was found for the other cell populations in our scRNA-seq data (Fig. 3k).

With the aim of gaining a broader understanding of biological processes potentially affected by gene expression changes in *Elmo2⁻/⁻* ECs and MSCs, a gene set enrichment analysis including all de-regulated genes with a (less stringent) *p*-adjusted cut-off value of 0.01 and a log2 fold change > 0.5 or < − 0.5 was performed. From the top gene ontology terms found (Supplementary Fig. 5g) there is no explicit relation to blood vessel development either in the ECs or MSCs population, yet different aspects related to extracellular matrix organization are highlighted for both cell types. Thus, unexpectedly, the analysis of the scRNA-seq data reveals rather limited changes in gene expression and provides no clear explanation for the dramatic changes in the mutant common carotid arteries.

## Inactivation of *Elmo2* in endothelial and smooth muscle cells

For cell type-specific loss-of-function experiments, a conditional (loxP-flanked) allele of *Elmo2* (Supplementary Fig. 1a) was established and validated by breeding it to homozygosity in a *PGK-Cre⁺/ᵀ* background[43]. As expected, ubiquitous and constitutive Cre activity in these mutants led to widespread *Elmo2* inactivation and phenocopied the vascular defects seen *Elmo2⁻/⁻* embryos generated with the "knockout-first" approach (Supplementary Fig. 6a–d). Next, we generated EC-specific mutants by interbreeding of mice carrying the loxP-flanked *Elmo2* allele and *Tek-Cre* transgenic animals[44]. Analysis with the *R26-mTmG* Cre-reporter[45] confirmed successful *Tek-Cre*-mediated recombination in the embryonic endothelium (Supplementary Fig. 7a). However, EC-specific *Elmo2* mutants (*Elmo2ᐃEC*) showed no observable defects in vascular development (Supplementary Fig. 7b). In particular, the size and morphology of *Elmo2ᐃEC* carotid arteries (Supplementary Fig. 7b, c) as well as the expression of known EC or VSMC markers (Supplementary Fig. 7d) are indistinguishable from control littermates. These results argue that the vascular malformations observed in global *Elmo2* knockout embryos are not caused by cell-autonomous defects in the endothelium.

Next, we conducted genetic experiments to address whether *Elmo2* is required in VSMCs. To this end, a transgene expressing constitutive Cre under the transcriptional control of Transgelin (SM22α) (*Tagln-Cre*[46]) was introduced into the *Elmo2* conditional (floxed) background, and embryos at specific developmental stages were collected. Unexpectedly, the resulting smooth muscle cell-specific

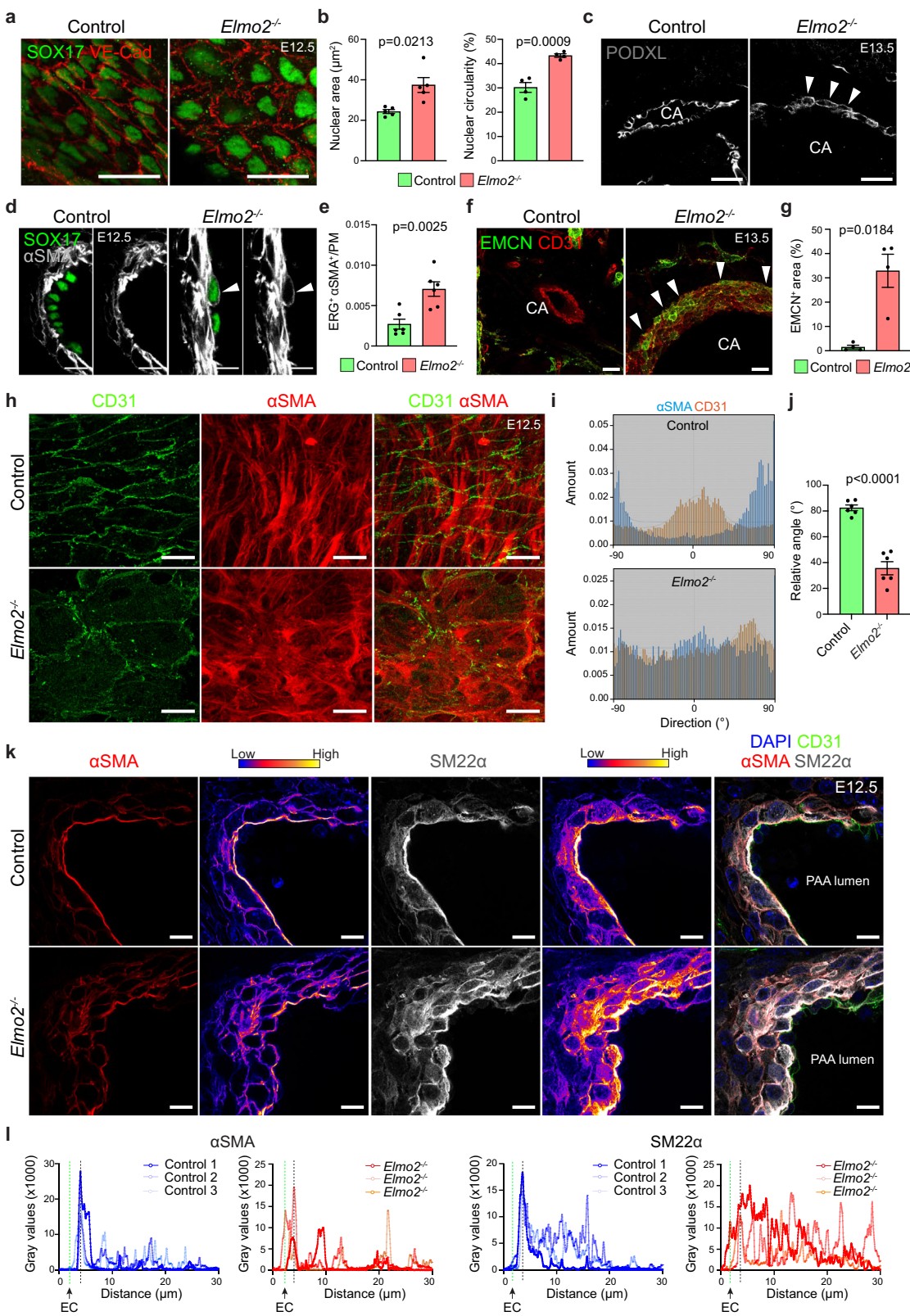

knockout embryos (*Elmo2*ᐧᐧᵃSMC) were macroscopically indistinguishable from control littermates at E13.5 (Fig. 4a). Immunostaining-assisted analysis of histological sections revealed subtle, yet statistically significant dilation of the carotid arteries at E13.5 and E15.5 (Fig. 4b, c and Supplementary Fig. 8a). Despite the carotid artery dilation, other relevant features of the global knockout phenotype were not reproduced after *Tagln-Cre*-mediated deletion of *Elmo2*. In particular, there

were no signs of aneurysm formation, disorganized VSMC alignment, or altered endothelial morphology, polarity, and gene expression (Fig. 4d and Supplementary Fig. 8b).

To rule out that the failure to reproduce the global knockout phenotype is caused by suboptimal *Elmo2* deletion in VSMCs, the recombination efficiency of *Tagln-Cre* in E13.5 embryos was assessed with the *R26-mTmG* Cre-reporter. Analysis of GFP expression as a

**Fig. 2 | Loss of *Elmo2* causes defects in the third pharyngeal arch (3rd PAA) and carotid arteries (CAs).** **a** Confocal images of the 3rd PAA from E12.5 embryos stained for endothelial cell (EC) junctions (VE-Cadherin, red) and arterial EC nuclei (SOX17, green). Scale bars, 10 μm. **b** EC nuclear area (left) and circularity (right) in the 3rd PAA from E12.5 embryos. Mean ± SEM, $n = 5$ and Welch's *t*-test (area), $n = 4$ and unpaired *t*-test (circularity). **c** Carotid artery (CA) cross-sections stained with the EC-luminal marker Podocalyxin (PODXL, gray) showing abluminal mislocalization (white arrowheads) in *Elmo2*-/- E13.5 embryos. Scale bars, 25 μm. **d** 3rd PAA stained for arterial EC nuclei (SOX17, green) and vascular smooth muscle cells (VSMCs, αSMA, gray) showing ectopic expression of αSMA in ECs (white arrowheads) from E12.5 *Elmo2*-/- embryos. Scale bars, 10 μm. **e** Relative number of ERG+ ECs with ectopic expression of αSMA normalized to vessel perimeter (PM) in the 3rd PAA from E12.5 embryos. Mean ± SEM, $n = 6$. Unpaired *t*-test. **f, g** Confocal images (**f**) and corresponding quantitation (**g**) reflecting persistent expression of endomucin

(EMCN, green) in arterial ECs (CD31, red) of E13.5 *Elmo2*-/- CAs (white arrowheads). Scale bars, 25 μm. Mean ± SEM, n = 4. Welch's *t*-test (**g**). **h** Super-resolution confocal images of the 3rd PAA from E12.5 embryos stained for ECs (CD31, green) and VSMCs (αSMA, red). Scale bars, 10 μm. **i, j** Graphical representation (**i**) and corresponding quantitative analysis (**j**) of VSMC-actin bundles' (αSMA, blue) alignment with respect to ECs' longest axis (CD31, orange) in E12.5 embryos. Data in (**j**) represented as Mean ± SEM, n = 6. Unpaired *t*-test. **k** Confocal images of 3rd PAA cross-sections stained for VSMC markers (αSMA, red or SM22α, gray) and color-coded representation of signal intensity in E12.5 embryos. Panel on the right shows merged channels including staining for nuclei (DAPI, blue) and ECs (CD31, green). Scale bars, 10 μm. **l** Intensity profile of αSMA and SM22α immunosignals from 3rd PAA cross-sections of E12.5 embryos. The green dashed line indicates the position of ECs (ectopic expression of αSMA and SM22α in *Elmo2*-/-). The black dashed line indicates the position of the first VSMCs.

surrogate marker of recombination clearly showed that the vast majority of VSMCs and mesenchymal cells around the carotid arteries (Fig. 4e) and neighboring vessels (Supplementary Fig. 8c) are efficiently targeted by the *Tagln-Cre*. In summary, these results argue that *Elmo2* deletion in VSMCs by means of a Transgelin-driven constitutive Cre-recombinase is not able to fully phenocopy the effects elicited upon global gene inactivation.

### Neural crest-specific deletion of *Elmo2* phenocopies the global *Elmo2* KO

Neural crest cells contribute to many craniofacial tissues and are an important source of mural cells in the 3rd PAA and thereby the common carotid arteries[35,47]. We chose *Wnt1-Cre2* transgenic mice[48] to study NCCs and their progeny, which would also address potential roles of *Elmo2* early in mural cell differentiation, whereas *Tagln-Cre* targets more differentiated VSMCs.

In order to assess if *Wnt1-Cre2* allows targeting of neural crest-derived VSMCs progenitors and compare the recombination timing with that of *Tagln-Cre*, lineage tracing analysis using the *R26-mTmG* reporter allele was carried out for both lines. Notably, *Wnt1-Cre2*-mediated labeling allows detection of a large number of GFP+ cells that cluster around the vascular plexus giving rise to the 3rd PAA by E9.5, whereas *Tagln*-Cre-mediated recombination is restricted to the heart at the same stage (Supplementary Fig. 9a). In line with this result, expression of the VSMC markers αSMA and SM22α is limited to the heart and cannot be detected in mural cells around the blood vessels in the branchial arches (Supplementary Fig. 9b). Expression of SM22α in mural cells or the 3rd PAA is first detected at E10.5, which coincides with the emergence of *Tagln-Cre*-labeled GFP+ cells in this structure (Supplementary Fig. 9c). In contrast, at this timepoint, a much larger number of *Wnt1-Cre2*-traced cells wrap around the 3rd PAA forming a surrounding layer that consists of both SM22α+ VSMCs and as yet undifferentiated cells (Supplementary Fig. 9c). At E11.5, recombination with both Cre lines generates robust perivascular GFP labeling in the relevant region (Supplementary Fig. 9d).

We further analyzed the recombination pattern elicited by *Wnt1-Cre2* in E13.5 embryos with special interest to the carotid arteries and neighboring vascular structures. *Wnt1-Cre2*-mediated recombination targeted VSMCs around the carotid arteries with high efficacy but, as expected, spared the mesoderm-derived smooth muscle surrounding the vertebral arteries, dorsal aorta, and jugular veins, as well as the endothelial lining of blood vessels (Fig. 5a and Supplementary Fig. 10a–c). Taken together, these results prove that *Wnt1-Cre2* efficiently targets NCC-derived VSMCs and their progenitors in the early mouse embryo.

Remarkably, neural crest-specific mutants (*Elmo2*ΔNCC), generated by interbreeding of the *Elmo2* conditional line with *Wnt1-Cre2*, reproduce the vascular defects observed in the global knockout model. Macroscopic observation of E13.5 *Elmo2*ΔNCC embryos revealed severe hemorrhaging in the cervical region and dorsal edema (Fig. 5b) as well

as massive dilation of the carotid arteries with aneurysm formation (Fig. 5c, d). Moreover, the defects in the hierarchical expression pattern of αSMA and SM22α found in *Elmo2*-/- embryos are also present upon neural crest-specific deletion (Fig. 5e), as well as the disorganized alignment of αSMA+ actin bundles with respect to the longest EC axis (Fig. 5f, g).

Strikingly, phenotypic changes observed in the ECs of the global knockout, such as retained expression of Endomucin in arterial territories, discontinuous endothelial lining, and ectopic expression of mesenchymal markers, were also seen upon neural crest-specific *Elmo2* deletion (Fig. 5h, i). Given that *Wnt1-Cre2*-driven recombination spares the endothelium, this result is further evidence that the EC defects in *Elmo2* mutants are secondary and probably a consequence of the severe vessel dilation.

Since neural crest cells give rise to parts of the autonomic nervous system and might therefore control vascular tone through VSMC innervation[49], we interbred the *Elmo2* conditional knockout mice with the *TH-IRES-Cre* line[50], which directs Cre recombination to catecholaminergic sympathetic neurons. Efficient and precise targeting of the paravertebral sympathetic ganglia was confirmed by GFP expression in the *R26-mTmG* Cre reporter background (Supplementary Fig. 11a, b). Nevertheless, *Elmo2* deletion in these structures did not induce relevant phenotypic alterations (Supplementary Fig. 11b–d). The resulting mutants survived to term and were obtained slightly above the expected ratio at birth (27.3% instead of 25%). Altogether, these results indicate that *Elmo2* is essential for vascular diameter control in the developing carotid artery by regulating the properties of NCC-derived VSMC progenitors.

### *ELMO2* controls contractile ability and actin dynamics of human VSMCs in vitro

To gain insight into the cellular function of ELMO2, we conducted experiments in cultured VSMCs. We opted for human brain vascular smooth muscle cells (HBVSMCs) because these cells, just like those wrapping around the 3rd PAA and carotid artery, are of neural crest origin[51]. Moreover, HBVSMCs express -1000-fold higher levels of *ELMO2* compared to *ELMO1* (Supplementary Fig. 12a).

Silencing RNA (siRNA)-mediated knockdown (KD) of *ELMO2* in HBVSMCs significantly decreased transcript abundance already by 24 h after treatment and this effect was maintained over several days (Fig. 6a). At the protein level, significant reduction of ELMO2 was obvious at 48 h after siRNA treatment, with the highest depletion achieved at 72–96 h (Fig. 6b), suggesting a slow protein turnover rate. In vitro, *siELMO2*-treatment induced a compensatory ~2-fold increase in *ELMO1* transcription, which coincides with the timepoints of highest ELMO2 protein depletion (Supplementary Fig. 12b), yet this upregulation may be of limited functional significance given the much higher endogenous expression of *ELMO2* in HBVSMCs.

Similar to our in vivo observations, *siELMO2*-treated cells did not show relevant changes in the expression of known VSMC markers

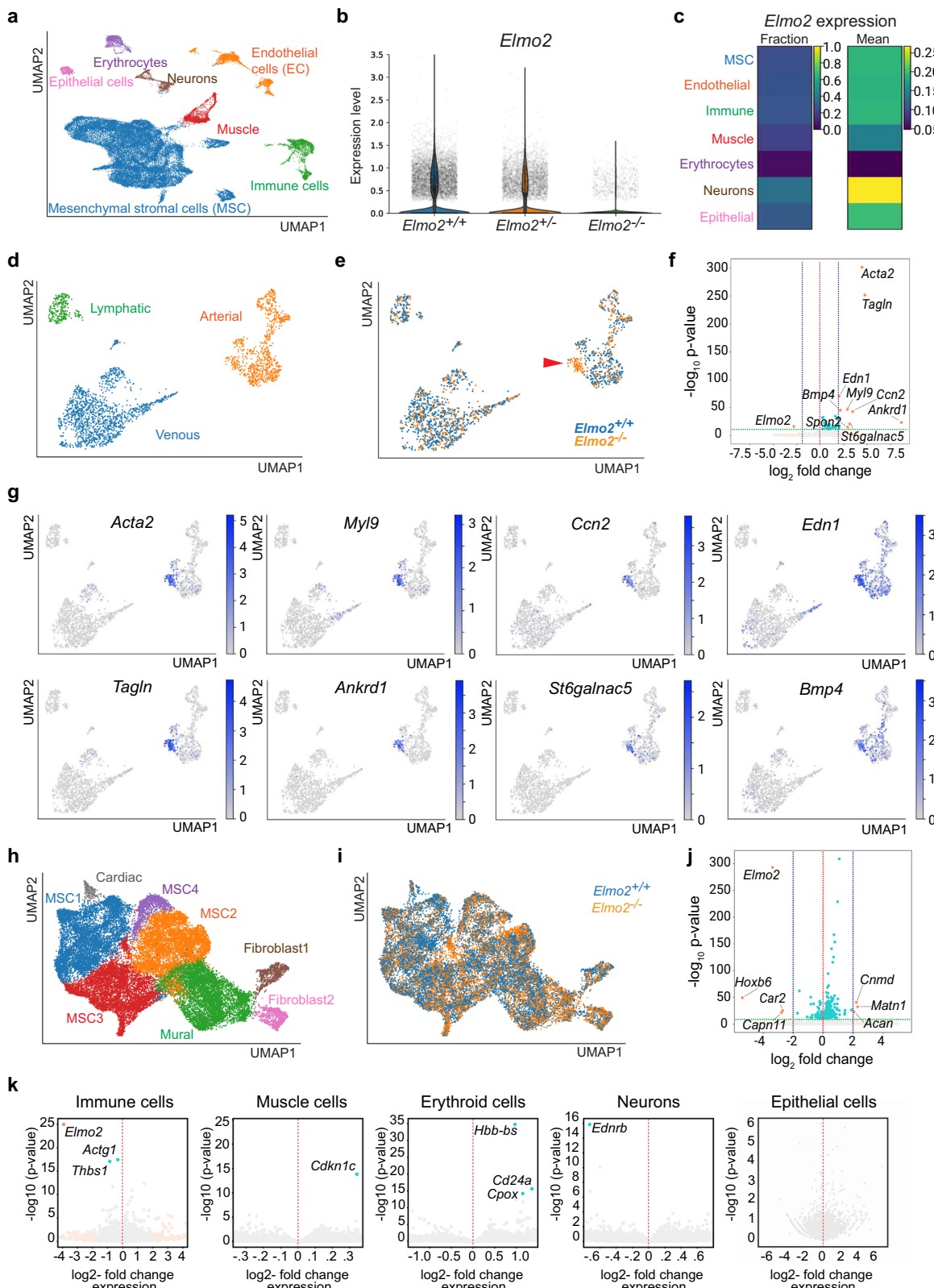

relative to *siControl* cells (Supplementary Fig. 12c, d). Likewise, the overall abundance of previously described ELMO2 interaction partners or downstream effectors relevant for cell adhesion, extracellular matrix binding and cell contractility were unchanged at the protein level (Supplementary Fig. 12e). The subcellular localization of integrin-linked kinase, a well described interactor of ELMO2, and the phosphorylation of myosin light chain 2 and 9, which are key regulators of

cell contractility, were comparable in KD and control cells (Supplementary Fig. 12d, f). However, analysis of F-actin by phalloidin staining (Supplementary Fig. 12d) revealed a slight reduction in the intensity and abundance of stress fibers in *siELMO2* HBVSMCs, whereas cortical actin appeared unaffected.

Next, we analyzed the functional performance of *siELMO2* HBVSMCs in assays requiring active remodeling of the cytoskeleton.

**Fig. 3 | scRNA-seq analysis of the E12.5 cervical region. a** Uniform manifold approximation and projection (UMAP) plot of different cell types in the cervical region of E12.5 mouse embryos. **b** Violin plot showing average expression level of *Elmo2* transcript in cells from the different samples analyzed. **c** Aggregate heat-maps showing the fraction of cells expressing *Elmo2* and the mean expression of *Elmo2* in the different cell types identified. Color represents a scaled fraction or expression level. **d** UMAP plot of the endothelial cell population in the cervical region of E12.5 mouse embryos. Colors represent different cell subclusters. **e** UMAP plot of the endothelial cell population in the cervical region of E12.5 mouse embryos. Colors represent different samples (*Elmo2*<sup>+/+</sup>, blue; *Elmo2*<sup>−/−</sup>, orange). The red arrowhead indicates a group of cells within the arterial population which is only present in the knockout embryos. **f** Volcano plot of differentially expressed genes between endothelial cells from control and *Elmo2*<sup>−/−</sup> E12.5 embryos. Blue dots, *p*-adjusted value <1$^{-10}$; orange dots, *p*-adjusted value <1$^{-10}$ and log$_2$ fold change > 2.0 or < − 2.0. Wald test and multiple testing correction by independent hypothesis

weighting (IHW). **g** UMAP plots depicting the expression pattern of selected genes, which appear upregulated in endothelial cells from *Elmo2*<sup>−/−</sup>. Color represents the scaled expression level. **h** UMAP plot of the different cellular subclusters identified within the mesenchymal stromal cell (MSC) population. **i** Color-coded UMAP plot of the MSC cluster comparing control (*Elmo2*<sup>+/+</sup>, blue) and knockout (*Elmo2*<sup>−/−</sup>, orange) cell populations. **j** Volcano plot of differentially expressed genes between MSCs from control and *Elmo2*<sup>−/−</sup> E12.5 embryos. Blue dots, *p*-adjusted value <1$^{-10}$; orange dots, *p*-adjusted value <1$^{-10}$ and log$_2$ fold change > 2.0 or < − 2.0. Wald test and multiple testing correction by independent hypothesis weighting (IHW). **k** Volcano plots showing differentially expressed genes in the indicated cell populations after comparison of control (*Elmo2*<sup>+/+</sup>) and *Elmo2*<sup>−/−</sup> E12.5 samples. Blue dots, *p*-adjusted value <1$^{-10}$; orange dots, *p*-adjusted value <1$^{-10}$ and log$_2$ fold change > 2.0 or < − 2.0. Wald test and multiple testing correction by independent hypothesis weighting (IHW).

Loss of ELMO2 reduced cell attachment and spreading in Collagen I-coated culture plates at early timepoints (10 min to 2 h) relative to *siControl* cells. Yet, these defects were no longer detectable at 6 h after seeding, when both the area and number of cells attached are indistinguishable between the KD and control conditions (Fig. 6c, d and Supplementary Fig. 12g). Time-lapse video recordings from live-imaging co-culture experiments confirmed the delayed adhesion of *siELMO2* cells (Supplementary Fig. 12h, i). Contrary to control cells, which extend filopodia-like cytoplasmic projections, spherical pro-trusions (blebs) were continuously formed and retracted in the membrane of KD cells (Fig. 6e and Supplementary Movie 1). More-over, culture of *siELMO2* HBVSMCs in 3D fibrin hydrogels led to a similar reduction in cell spreading and cellular area relative to *siControl* cells, consistent with the results seen in 2D experiments (Fig. 6f, g).

In order to test the contractile capacity of VSMCs in vitro, we treated co-cultured control and KD cells with carbachol, a cholinergic agonist that increases cytoplasmic calcium levels and stimulates the RhoA/ROCK (Rho-associated kinase) pathway[52]. Live-imaging analysis showed significantly impaired contractility of *ELMO2* KD cells, which were not able to efficiently retract their cellular projections upon carbachol treatment (Fig. 6h, i and Supplementary Fig. 12j). Further verifying this observation, 3D collagen gel contraction assays confirmed that *siELMO2*-treatment drastically impaired the contractile capacity of HBVSMCs (Fig. 6j, k). These results point out to an important role for ELMO2 in the regulation of HBVSMC actin dynamics in the context of cell adhesion, spreading, and contraction.

With the aim of uncovering the reasons behind the deficient regulation of actin dynamics after *ELMO2* downregulation, the ratio of globular (G)-actin to filamentous (F)-actin was determined as a readout of actin polymerization. F-actin abundance was found to be sig-nificantly decreased in *siELMO2* HBVSMCs relative to *siControl* cells (Fig. 6l), which showed a higher F-actin to G-actin ratio (Fig. 6m). These changes in the functional state of the actin cytoskeleton may be, at least in part, a consequence of reduced active Rac1 in *ELMO2* KD cells, as shown by a G-LISA-based activation assay (Fig. 6n).

The reduced abundance of F-actin prompted us to test the impact of pharmacological actin filament stabilization. For this, *siControl* and *siELMO2* HBVSMCs were analyzed after treatment with jasplakinolide (JAS), a cyclic peptide known to bind and stabilize filamentous actin in vitro[53]. Notably, JAS allowed efficient formation of filopodia-like structures and improved adhesion and spreading of *siELMO2* cells to an extent that was undistinguishable from control cells (Fig. 6o, p and Supplementary Fig. 12k). Likewise, *siELMO2* cells recovered their con-tractile ability after JAS treatment in the collagen gel contraction assays (Fig. 6q, r). Altogether, these data indicate that promoting actin polymerization and stabilization restores functional features of ELMO2-deficient HBVSMCs.

## Timing of global *Elmo2* deletion is determinant for aneurysm formation

Our in vitro data point to contractility defects as the most likely cause for the vascular dilation phenotype in vivo. Yet, the most severe phe-notypic alterations are only triggered upon *Elmo2* deletion in neural crest-derived progenitors (*Wnt1-Cre2*-mediated recombination) and do not reach the same extent when contractile VSMCs are targeted (through *Tagln-Cre*). In this regard, the early onset of *Wnt1-Cre2*-mediated recombination in the 3<sup>rd</sup> PAA (Supplementary Fig. 9) and the mild phenotype of *Tagln-Cre*-generated *Elmo2* mutants raise the pos-sibility that the gene product is required during an early stage of vessel wall assembly.

In order to directly assess this hypothesis, we bred the *Elmo2* conditional knockout model with mice expressing an inducible recombinase (CreERT2) under control of the ubiquitously expressed *Rosa26* locus[54] (*R26-CreERT2*). Pregnant dams were treated with 4-hydroxytamoxifen (4-OHT) at defined stages of embryonic devel-opment in order to resemble the recombination timing achieved with *Wnt1-Cre2* and *Tagln-Cre* lines.

4-OHT treatment at E8.5 and E9.5, which coincides with the timing of *Wnt1-Cre2* activity, induced severe vascular defects, including car-otid artery aneurysm by E13.5 (Fig. 7a–c). These defects are identical to those observed upon global or NCC-specific *Elmo2* inactivation and correspond to a complete depletion of ELMO2 protein (Fig. 7d). In contrast, 4-OHT administration from E10.5 to E11.5, mimicking the later recombination by *Tagln-Cre*, did not lead to aneurysm formation by E13.5 but induced a milder dilation of the carotid arteries (Fig. 7e–g). Given that the short time period between 4-OHT treatment and ana-lysis in this treatment regime results in incomplete depletion of ELMO2 protein (Fig. 7h), we tested whether a longer waiting interval could increase the severity of the resulting phenotype. Interestingly, 4-OHT administration from E10.5 to E11.5 with embryo collection at E15.5 (Fig. 7i–k) did not aggravate the arterial dilation despite efficient ELMO2 reduction (Fig. 7l).

These results indicate that the ELMO2 function is essential in the 3<sup>rd</sup> PAA in the early embryo, but no longer required for carotid artery development in the second half of gestation. The early emergence of these defects in global and NCC-specific mutants establishes that ELMO2 is directly involved in the regulation of vascular morphogen-esis, which argues that vascular defects are probably not secondary to skeletal overgrowth in VMOS patients. Furthermore, it is striking that ELMO2 is specifically required in the vessels supplying the mandible and maxilla, which raises the possibility that the defects in these ske-letal elements are a consequence of the vascular dilation.

## Discussion

ELMO family proteins and, in particular, the *C. elegans* ortholog CED-12, were initially identified as regulators of phagocytosis and cell migration, which act in concert with Dock family guanine nucleotide

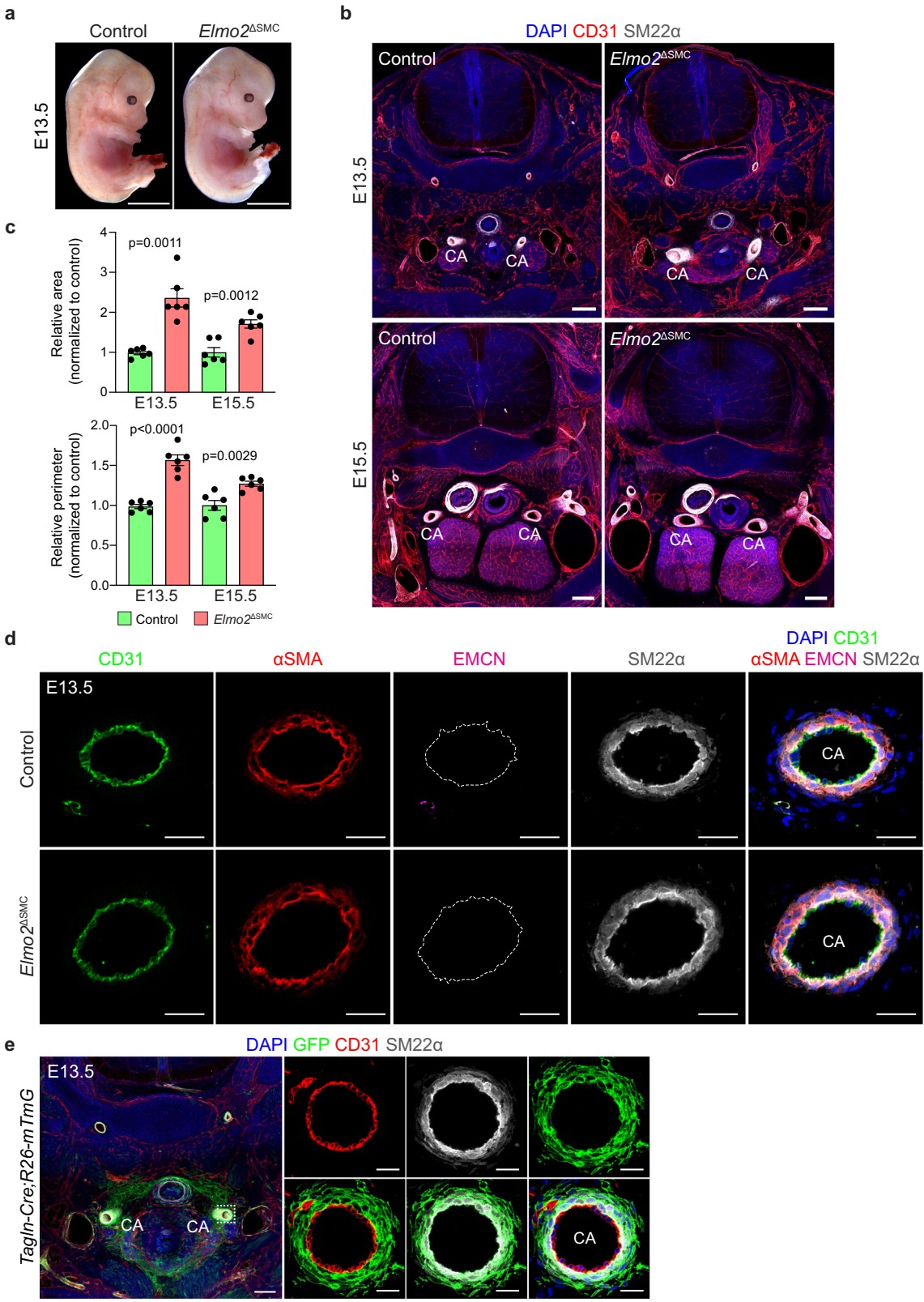

exchange factors (GEFs) and the Rac1 small GTPase[12,55–57]. Later studies have added integrin-linked kinase as an interaction partner of ELMO with relevance for processes such as endosome trafficking, actin remodeling, and the regulation of microtubule dynamics[15,21,58]. While all three mammalian ELMO family members share a similar domain architecture and are established interaction partners of DOCK family members that regulate Rac1 activity[59,60], mice carrying a global

deletion of *Elmo1* or *Elmo3* are viable, whereas the inactivation of *Elmo2* leads to lethality after midgestation[17]. The latter, as we show here, is a consequence of the essential role of ELMO2 in the 3rd PAA and common carotid artery development.

Despite of massive vessel dilation and endothelial defects that probably reflect overstretching of the *Elmo2* mutant arterial wall, alterations in gene expression, detected by scRNA-seq analysis, are

**Fig. 4 | Deletion of *Elmo2* using *Tagln-Cre* leads to subtle carotid artery dilation.** **a** Representative images of control and *Elmo2*$^{\Delta SMC}$ E13.5 embryos showing no macroscopic defects upon *Elmo2* deletion in vascular smooth muscle cells (VSMCs). Scale bars, 2 mm. **b** Confocal overview images of transverse sections from control and *Elmo2*$^{\Delta SMC}$ embryos stained for nuclei (DAPI, blue), endothelial cells (ECs, CD31, red), and VSMCs (SM22α, gray) showing mild dilation of the carotid artery (CA) in E13.5 and E15.5 mutant embryos. Scale bars, 200 μm. **c** Quantitation of area and perimeter of the carotid arteries in E13.5 and E15.5 *Elmo2*$^{\Delta SMC}$ embryos normalized to control (set as 1). Mean ± SEM, *n* = 6. Welch's *t*-test (area E13.5) and unpaired *t*-test (area E15.5 and perimeter at all stages). **d** Confocal images of carotid arteries (CA) from E13.5 control and *Elmo2*$^{\Delta SMC}$ embryos stained for nuclei (DAPI, blue), ECs (CD31, green), veins/capillaries (EMCN, magenta), and VSMCs (αSMA, red; SM22α, gray). Scale bars, 25 μm. **e** Recombination analysis in *Tagln-Cre;R26-mTmG* E13.5 embryos. Representative confocal images of the carotid artery (CA, white dashed line square in overview image) stained for nuclei (DAPI, blue), recombined cells (GFP, green), ECs (CD31, red), and VSMCs (SM22α, gray). Higher magnification images show GFP expression restricted to VSMCs and surrounding mesenchymal cells. Scale bars, 200 μm (overview) and 20 μm (higher magnification).

unexpectedly minor. This applies to mutant ECs but also to MSCs, the largest cell cluster in our scRNA-seq data, which also encompasses different populations of mural cells. The transcriptomic results are consistent with our phenotypic characterization of mutants, which shows that ELMO2 is not necessary for NCC migration into the pharyngeal arches nor for VSMC specification, as indicated by persisting expression of key markers in mutant cells. Accordingly, it should be considered whether the reported reduction or absence of VSMCs inside VMOS lesions[2] might be a consequence of vessel dilation and not a primary defect caused by ELMO2 loss-of-function mutations.

Our in vitro experiments also support that ELMO2 is not controlling VSMC identity or survival but rather regulates cell contractility and spreading through Rac1 and the modulation of actin dynamics. In this context, it is worth noting that NCC-specific deletion of ILK and Rac1, two critical interaction partners of ELMO2 and well-established regulators of cytoskeletal dynamics, leads to impaired PAA development[29,61]. Neural crest cell migration, however, is not compromised in these mutants, similar to what we report for *Elmo2*-deficient embryos. Interestingly, inactivation of *Ilk* in NCCs impairs VSMC differentiation inside branchial arches[61], which might reflect ELMO2-independent roles of ILK in cell adhesion and specification. On the other hand, NCC-specific deletion of Rac1 results in aberrant patterning of pharyngeal arch arteries, defective outflow tract septation, and aneurysms in the vessels branching from the common arterial trunk, without impairing VSMC specification[29]. These phenotypic similarities argue that ELMO2 might act in concert with Rac1 and ILK, consistent with previous reports[9,13,14].

Given that the ELMO2 function appears to be critically required for the contractile and vessel-stabilizing function of NCC-derived smooth muscle cells surrounding the 3rd PAA, the absence of defects in other vascular structures is highly surprising. Strikingly, even the neighboring *Elmo2*$^{-/-}$ PAAs remain normal or are affected to a very low degree. Although these differences cannot be explained by the ontogeny of the vascular components, as endothelial and mural cells of all PAAs are derived from the secondary heart field (SHF)[62] and the neural crest[47,63–65], respectively, we cannot rule out that distinct embryonic origins in combination with local factors or mechanical forces play a role in the specificity of the vascular lesions. Of particular interest, vessel dilation in VMOS patients directly affects the external carotid arteries, whereas bilateral ophthalmic internal carotid artery aneurysms have also been reported[4], suggesting that potentially overlapping mechanisms may be responsible for the increased susceptibility of these vessels in humans and the 3rd PAA in mice. Redundant activity of other ELMO family members in unaffected vessels might be one explanation and future studies involving compound mutants might be able to address this question.

During embryonic development, the PAAs undergo large-scale asymmetric morphogenesis to form critical structures of the aorta-associated arterial network. The extensive transformation of the pharyngeal arch vessels occurs within a dynamic biomechanical environment that is fundamentally influenced by blood flow and hemodynamic forces. Quantitative analysis of blood flow, velocity, and wall pressure in avian embryos has shown that the 3rd PAA receives the largest amount of flow during stages that are equivalent to E10.5-E11.5

of mouse development[66]. In addition, it needs to be considered that the 3rd PAAs persist and significantly contribute to the formation of the common carotid arteries, whereas the other PAAs partially or completely regress, which implies that their contribution to the adult vasculature is comparably minor[37]. These may be important factors explaining the locally restricted phenotypic alterations.

With regard to VMOS, our work establishes that vascular malformations emerge before the development of the craniofacial skeleton and therefore independently from bone defects. This finding is important because osteoblast lineage cells are a source of vascular endothelial growth factor A (VEGF-A), a master regulator of blood vessel growth and patterning, which might trigger vessel dilation and increased permeability in settings of bone overgrowth[67,68]. Thus, while the analysis of VMOS patients is usually confined to subjects with fully established lesions, more attention should be given to the detection of vascular alterations during the onset and early development of the disease. In addition, our findings raise the possibility that carotid artery defects, altered blood flow, and increased vascular permeability may be potential underlying causes of craniofacial bone overgrowth. This might also explain the strong involvement of the mandible and maxilla, which are supplied by arteries, namely the maxillary and facial arteries, that branch off from the external carotid arteries.

The embryonic lethality of global and NCC-specific *Elmo2* mutants currently precludes a direct investigation of the interplay between bone growth and the side branches of the carotid artery system. Moreover, there are limitations for experimental approaches involving inducible Cre recombinase systems because, as our results with *R26-CreERT2* mice show, timing is critical and only early activation of Cre activity will induce the full range of defects in the 3rd PAA and common carotid artery.

Human VMOS patients appear unaffected at birth and exhibit their first symptoms during childhood, which progressively worsen during development to adulthood[3,69]. Future work should try to address whether mutations in human *ELMO2* could lead to partial lethality during embryonic development. It should be also considered whether the previously reported residual activity of mutated ELMO2 gene products[2] might facilitate fetal survival but cause milder vascular defects that can trigger bone overgrowth.

The sum of our in vivo findings (summarized in Supplementary Table 1) firmly establishes that ELMO2 is necessary for normal formation of the 3rd PAA and the common carotid arteries. Moreover, our results raise important new questions regarding the etiology of VMOS and the potential benefit of vessel-targeting therapies for the treatment of this devastating disease.

## Methods
### Animals models
Mice used in this study were housed in the animal facility of the Max Planck Institute for Molecular Biomedicine, Münster. Animals were kept in individually ventilated cages under controlled temperature (22 °C +/− 1.5 °C) and humidity (55% +/− 5%), with a 14 h light:10 h dark cycle, and *ad libitum* supply of food and water. All animal procedures were conducted in accordance with the guidelines of the Max Planck Institute and approved by the *Landesamt für Natur, Umwelt und*

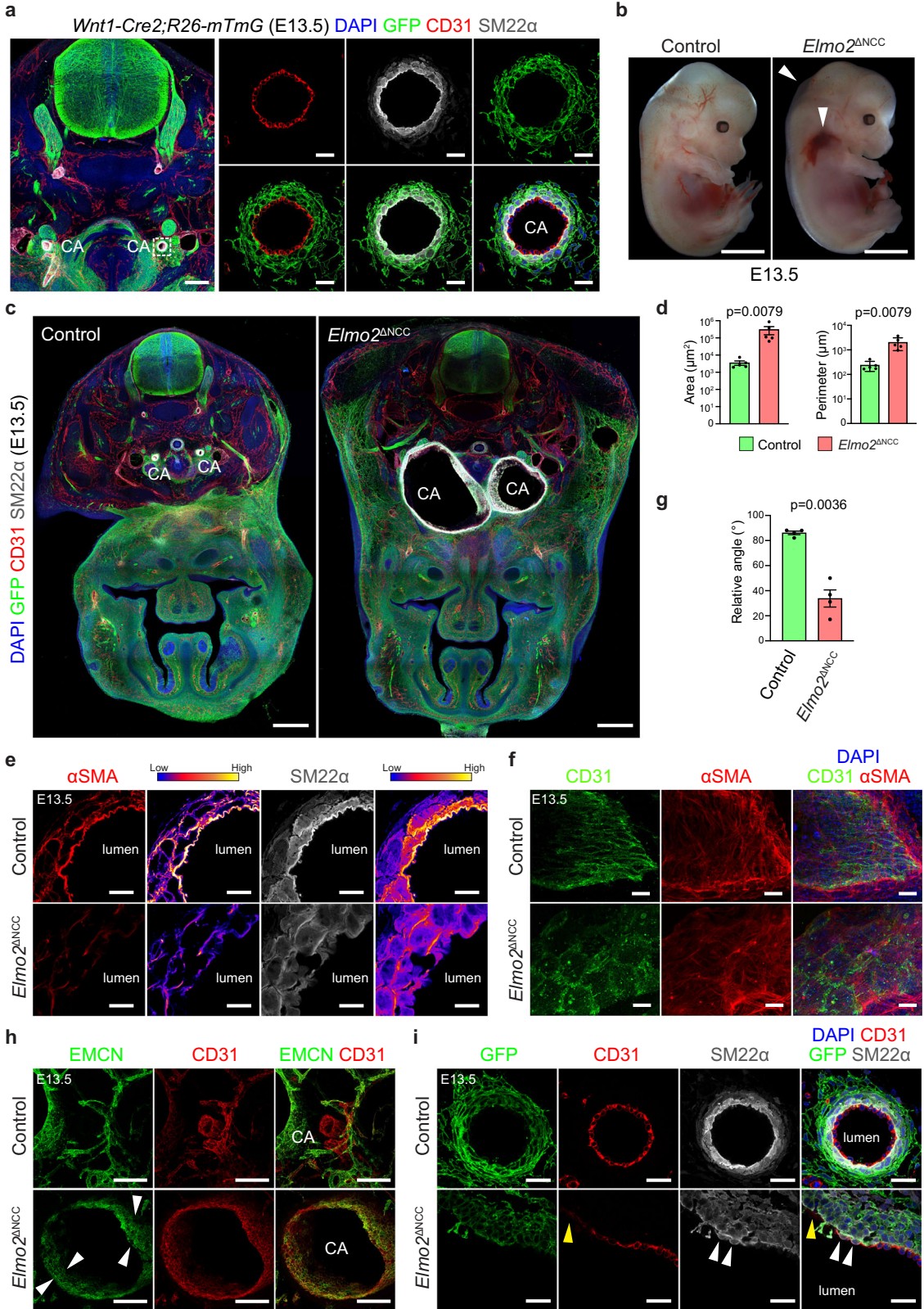

*Verbraucherschutz Nordrhein-Westfalen* (LANUV, Az. No. 81-02.04.2023.A383).

Mouse lines were maintained in a C57BL/6 J background. *Elmo2^LacZ* embryonic stem cells were obtained from the European Mouse Mutant Cell Repository (EMMCR- clone H08) and injected to blastocysts. Chimeras were selected based on coat color and genotype, and were further inbred to obtain *Elmo2^LacZ/+* animals. The *Elmo2^LacZ* allele has an internal ribosome entry site (IRES) sequence followed by the *LacZ* reporter gene and a polyadenylation signal (pA) upstream of the *Elmo2* exon 7. To generate *Elmo2* global knockout mice, female *Elmo2^LacZ* heterozygotes were bred to *PGK-Cre*[43] males. Offspring was backcrossed to wildtype animals, and Cre-negative

**Fig. 5 | Neural crest-specific deletion of *Elmo2* phenocopies the global *Elmo2* knockout. a** Recombination analysis of *Wnt1-Cre2;R26-mTmG* E13.5 embryos. Representative confocal images of the carotid artery (CA, white dashed line square in overview image) stained for nuclei (DAPI, blue), recombined cells (GFP, green), endothelial cells (ECs, CD31, red), and vascular smooth muscle cells (VSMCs, SM22α, gray). Scale bars, 200 μm (overview) and 25 μm (higher magnification). **b** Representative images of control and *Elmo2*$^{\Delta NCC}$ E13.5 embryos showing hemorrhages in the cervical region and subcutaneous edema (white arrowheads) in the neural crest-specific knockout mice. Scale bars, 2 mm. **c** Confocal overview images of cross-sections from control and *Elmo2*$^{\Delta NCC}$ E13.5 embryos stained for nuclei (DAPI, blue), recombined cells (GFP, green), ECs (CD31, red), and VSMCs (SM22α, gray) showing aneurysm formation in the carotid artery (CA) of neural crest-specific knockout mice. Scale bars, 500 μm. **d** Quantitation of area and perimeter of the carotid arteries in control and *Elmo2*$^{\Delta NCC}$ E13.5 embryos. Mean ± SEM, n = 5. Mann-Whitney test. **e** Confocal images of the carotid artery from control and *Elmo2*$^{\Delta NCC}$ E13.5 embryos stained for VSMCs (SM22α, gray; αSMA, red) and color-coded representation of signal intensity (right panels). Scale bar, 10 μm. **f** High-resolution confocal images of carotid arteries stained for ECs (CD31, green), VSMCs (αSMA, red), and nuclei (DAPI, blue) showing aberrant alignment of αSMA-bundles with respect to the longest axis of ECs in E13.5 *Elmo2*$^{\Delta NCC}$ embryos. Scale bars, 10 μm. **g** Quantitation of average relative angle between the ECs' longest axis and VSMC-actin bundles in control and *Elmo2*$^{\Delta NCC}$ E13.5 embryos. Mean ± SEM, n = 4. Welch's *t*-test. **h** Confocal images of carotid arteries (CA) stained for veins/capillaries (EMCN, green) and ECs (CD31, red) showing persistent expression of endomucin in the CA (white arrowheads) of E13.5 *Elmo2*$^{\Delta NCC}$ embryos. Scale bars, 100 μm. **i** Confocal images of carotid arteries stained for nuclei (DAPI, blue), ECs (CD31, red), recombined cells (GFP, green) and VSMCs (SM22α, gray) showing abnormal expression of SM22α in ECs (white arrowheads) and discontinuities in the endothelial lining (yellow arrowhead) of E13.5 *Elmo2*$^{\Delta NCC}$ embryos. Scale bars, 25 μm.

progeny carrying one copy of the *Elmo2* knockout allele (*Elmo2*$^{+/-}$) was selected to maintain the line. For global deletion experiments, embryos were obtained by crossing heterozygous *Elmo2* knockout (*Elmo2*$^{+/-}$) males and females. The *Elmo2* floxed allele was generated by Flp-mediated excision of FRT flanked sequences in *Elmo2*$^{LacZ}$, resulting in loxP sites flanking exon 7. In order to drive conditional deletion of *Elmo2*, Cre-positive males from tissue-specific or tamoxifen-inducible mouse lines, bearing one copy of the Elmo2 floxed allele, were interbred with homozygous *Elmo2* floxed females. The following Cre lines were used for the conditional deletion of *Elmo2* in different cell types: *Tek-Cre*[44] for targeting the endothelium, *Tagln-Cre*[46] to induce recombination in committed VSMCs, *Wnt1-Cre2*[48] for deletion in neural crest cells, and *TH-IRES-Cre*[50] to drive recombination in neuronal subpopulations. For global tamoxifen-inducible recombination, the *R26-CreERT*[54] model was used, and the *R26-mTmG* reporter[45] was utilized to monitor the efficiency and specificity of the different Cre-drivers.

Embryos at specific developmental stages were obtained from timed matings. The day in which the vaginal plug is detected is considered as embryonic day (E) 0.5. For sample collection, pregnant females were sacrificed by $CO_2$, the whole uterus removed and transferred to ice-cold PBS. Each embryo was carefully dissected out from the amnion and kept in ice-cold PBS for macroscopic imaging (AxioObserver, Zeiss). For immunostaining approaches, embryos were fixed in 2% PFA overnight at 4 °C with gentle agitation after severing off the head. For biochemistry experiments, embryos were flash frozen in liquid nitrogen.

## Tamoxifen administration

4-hydroxytamoxifen (4OHT, Sigma, Cat#H7904) was dissolved in a 1:1 mixture of ethanol-Kolliphor EL and stored in 1 mg aliquots at −20 °C. On the day of administration, 1 mg 4OHT was thawed at 37 °C and diluted with 200 μL of prewarmed PBS. 1 mg of progesterone (Sigma, Cat#P3972) dissolved in 200 μL of a 1:10 mixture of ethanol-peanut oil was mixed with the 4OHT and administered to pregnant females by oral gavage on the specified days.

## Immunofluorescence and microscopy

Vibratome sections with a thickness of 100-200 μm were permeabilized and blocked by incubation in 0.5% Triton-X-100, 1% BSA dissolved in PBS (blocking buffer) at 4 °C for 48 h. A blocking buffer was used for dilution of primary and secondary antibodies, which were incubated at 4 °C for 48 h with gentle agitation. Sections were mounted with Fluoromount-G (Southern Biotech, Cat#0100-01) between two 60 mm coverslips separated by a 0.2 mm spacer. Whole-mount staining of early-stage embryos (E9.5 to E11.5) was performed in the same manner. Image acquisition was carried out using a confocal microscope (Zeiss, LSM880). Details of the primary and secondary antibodies used in this study are provided in Supplementary Table 2.

Whole-mount immunostaining and clearing of embryos from E11.5 to E13.5 was performed following the iDISCO protocol[70], and images were obtained using a light sheet microscope (M2Lasers, AURORA).

Super-resolution microscopy images were obtained with a Zeiss LSM880 confocal microscope equipped with an Airyscan module allowing improved spatial resolution and signal to noise ratio (https://pages.zeiss.com/rs/896-XMS-794/images/ZEISS-Microscopy_The-Basic-Principle-of-Airyscanning.pdf). All image stacks were acquired with Fast Airyscan mode using a Plan-Apochromat 63X oil objective (NA 1.4, Zeiss) with a voxel size of 0.04 μm − 0.04 μm − 0.19 μm and processed with Airyscan 3D processing function of ZEN Black (v2.31, Zeiss).

## Proliferation assay

For labeling of proliferating cells, a 10 mg/mL EdU solution was intraperitoneally injected to pregnant females (50 mg/Kg body weight) 2 h before sample collection. Embryos were fixed in 2% PFA and processed as previously described for immunostaining of vibratome sections. EdU$^+$ cells were detected using the Click-iT EdU Alexa-647 imaging kit (Invitrogen, cat. #C10340) following the manufacturer's instructions.

## X-gal staining

Embryos were fixed with X-gal fixative (1.5% paraformaldehyde, 0.2% glutaraldehyde, 5 mM EGTA (pH 8.0), 2 mM $MgCl_2$, and 1X PBS, dissolved in water) for 90 min at room temperature (RT) and washed 3 times with washing buffer (1X PBS, 2 mM $MgCl_2$, 0.001% sodium deoxycholate, 0.02% Nonidet P40 dissolved in water) for 30 min. Samples were stained in prewarmed X-gal solution (1 mg/mL X-gal (dissolved in dimethylformamide), 5 mM $K_3Fe(CN)_6$, 5 mM $K_4Fe(CN)_6$·$3H_2O$, dissolved in washing buffer) and incubated at 37 °C with gentle agitation. After overnight incubation, X-gal staining developed as the visible blue signal. Samples were washed with PBS for 3 times at RT and counterstained with Eosin.

## Single cell RNA sequencing

The 3$^{rd}$ PAA, together with the surrounding mesenchyme, was microdissected from freshly harvested E12.5 mouse embryos and immediately transferred to collection buffer (25 mM HEPES in DMEM) in a 48-well plate. Samples were transferred to 1.5 mL tubes containing 100 μL of pre-warmed digestion buffer (50 mg/mL Liberase TM and 0.1 mg/mL DNAse-1 in collection buffer) and incubated at 37 °C for 15 min with occasional mixing and cell disaggregation by pipetting with 200 μL sterile filter tips. Enzymatic activity was stopped by adding 1 mL of inactivation buffer (10% fetal calf serum in DMEM) followed by filtering the cell suspension through a 40 μm strainer. To increase the cell yield, the filter was further washed with 400 μL of collection buffer, and the flow-through was collected.

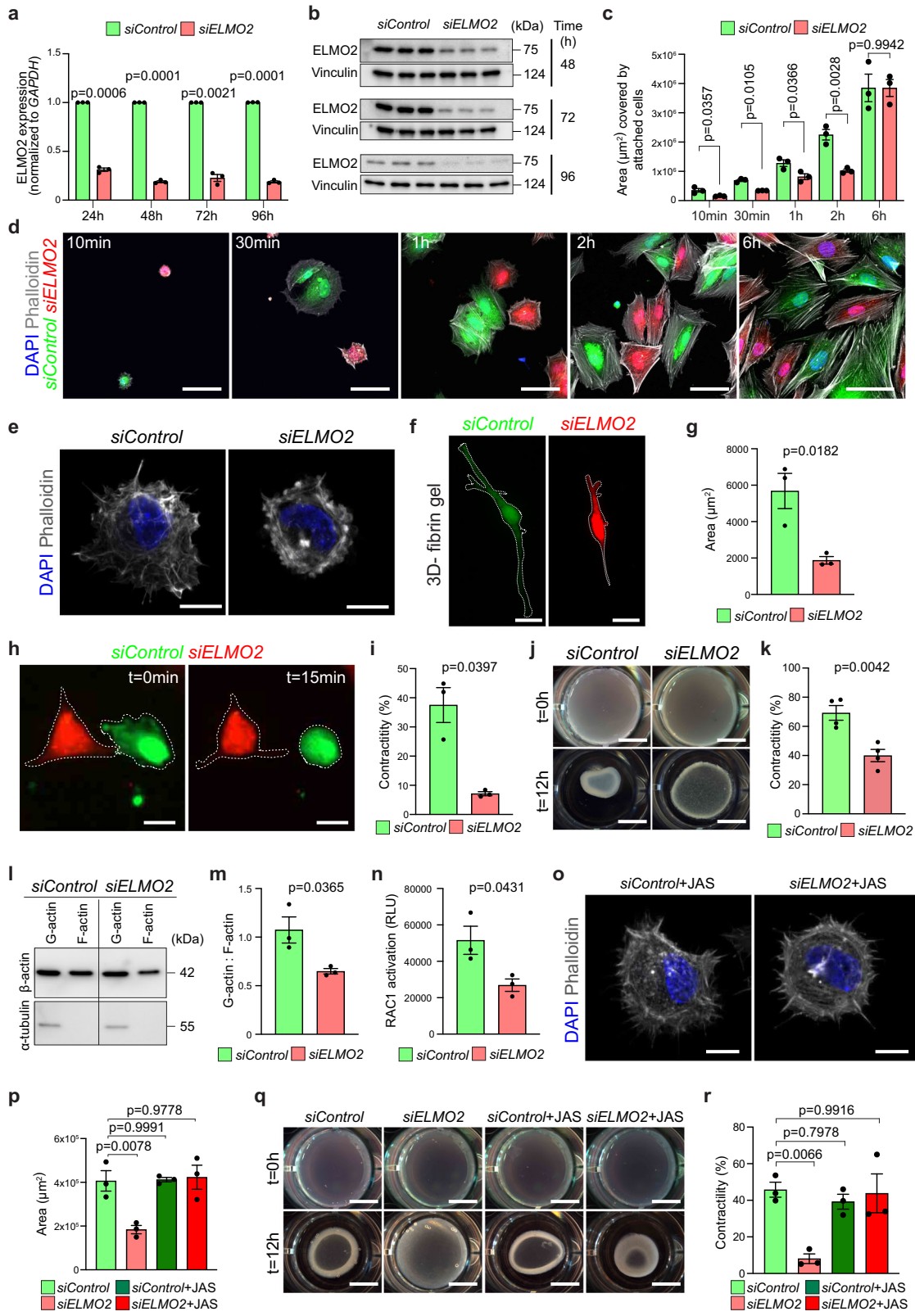

Samples were next centrifuged at $300 \times g$ and RT for 7 min. After careful aspiration of the supernatant, 100 μL of red blood cell-lysis buffer were used to resuspend the pellet. After a 1 min-RT incubation, 1.5 mL of FACS buffer (0.22 μm-filtered 2% heat-inactivated fetal calf serum in PBS) was added, and the samples were centrifuged ($300 \times g$ for 7 min at 4 °C) to pellet down the cells. Next, the supernatant was carefully aspirated and 100 μL of FACS buffer were used to gently

resuspend the cells. Cell yield was assessed with an automated cell counter (Logos Biosystems, LUNA-II™), and single cells were further captured with the BD Rhapsody Express Single-Cell Analysis System (BD Biosciences). Library preparation was performed following the manufacturer's instructions (BD Rhapsody™ system mRNA WTA library preparation protocol) and sequenced using Illumina NextSeq 500.

**Fig. 6 | Loss of *ELMO2* in human brain vascular smooth muscle cells (HBVSMCs) impairs actin dynamics and contractility. a** *ELMO2* expression (RT-qPCR) in *siControl* and *siELMO2* HBVSMCs at different timepoints (hours, h). Mean ± SEM, *n* = 3. Welch's *t*-test. **b** ELMO2 immunoblot from *siControl* and *siELMO2* HBVSMCs at different timepoints. Molecular weight markers (kDa) are indicated. **c, d** Area covered by attached HBVSMCs at different timepoints (**c**) and representative images (**d**) of *siControl* (green) and *siELMO2* (red) cells stained for nuclei (DAPI, blue) and actin (Phalloidin, gray). Mean ± SEM, *n* = 3. Welch's *t*-test (30 min) and unpaired *t* test (all other time points). Scale bars, 50 μm. **e** High magnification images of *siControl* and *siELMO2* HBVSMCs stained for nuclei (DAPI, blue) and actin (Phalloidin, gray) reveal membrane blebbing upon ELMO2-knockdown 10 min after seeding. Scale bars, 25 μm. **f, g** Representative images (**f**) and average cell area (**g**) of *siControl* (green) and *siELMO2* (red) HBVSMCs growing in a 3D fibrin gel. Scale bars, 50 μm. Mean ± SEM, *n* = 3. Unpaired *t*-test. **h, i** Representative images (**h**) and contractility analysis (**i**) of *siControl* (green) and *siELMO2* (red) HBVSMCs upon carbachol-treatment. Scale bars, 25 μm. Mean ± SEM, *n* = 3. Welch's *t*-test. **j, k** Representative images (**j**) and contractility analysis (**k**) of *siControl* and *siELMO2* HBVSMCs 12 h after seeding in collagen matrix. Scale bars, 5 mm. Mean ± SEM, *n* = 4. Unpaired *t*-test. **l, m** Immunoblot (**l**) and corresponding quantitation (**m**) of G-actin:F-actin ratio in *siControl* and *siELMO2* HBVSMCs. Molecular weight marker (kDa) is indicated. Mean ± SEM, *n* = 3. Unpaired *t* test. **n** Differences in active RAC1 in *siControl* and *siELMO2*-treated HBVSMCs. Mean ± SEM, *n* = 3. Unpaired *t*-test. **o** Representative images of *siControl* and *siELMO2* HBVSMCs treated with jaspla-kinolide (JAS) and stained for nuclei (DAPI, blue) and actin (Phalloidin, gray). Note rescued filopodia formation during cell spreading 10 min after seeding. Scale bars, 25 μm. **p** Area covered by attached *siControl* and *siELMO2* HBVSMCs after vehicle or jasplakinolide (JAS) treatment. Mean ± SEM, *n* = 3. Ordinary one-way ANOVA. **q, r** Collagen gel contraction assay (**q**) and corresponding quantitation (**r**) showing rescued contractility in jasplakinolide (JAS)-treated *siELMO2* HBVSMCs. Scale bars, 5 mm. Mean ± SEM, *n* = 3. Ordinary one-way ANOVA.

Raw FASTQ reads were quality and adapter trimmed using fastp (version 0.23.2, length cutoff 20, quality cutoff 15). The UMI and complex barcode were extracted and demultiplexed using custom scripts. The GRCm39 reference genome was merged with the reporter vector sequence to create a custom reference. STAR version 2.7.10a was used to generate a reference genome index, with reporter features and Gencode annotations vM29, subset to lncRNA and protein-coding genes. FASTQ reads were mapped against the reference genome index using STAR with the settings "--soloType CB_UMI_Simple --solo-CellFilter None --outSAMtype BAM SortedByCoordinate --soloFeatures GeneFull_Ex50pAS --soloCBstart 1 --soloCBlen 27 --soloUMIstart 28 --soloUMIlen 8 --soloCBwhitelist rhapsody_whitelist.txt --runRNGseed 1 --soloMultiMappers EM --readFilesCommand zcat". Raw counts were imported as AnnData objects. Low complexity barcodes were removed with the knee plot method, and cells with a mitochondrial mRNA content were further filtered out, as well as cells with unusually high total and gene counts, using manually determined cutoffs for each sample. Doublets were scored with Scrublet 0.2.3. Finally, each sample's gene expression matrix was normalized using Scran 1.22.1 with Leiden clustering input at resolution 0.5. G2M and S phase scores were assigned to each cell using gene lists from and the Scanpy 1.9.1 sc.tl.score_genes_cell_cycle function.

For embedding, clustering, and annotation, the normalized expression matrix was subset to the 3000 most highly variable genes (HVG, sc.pp.highly_variable_genes, flavor "seurat"). The top 100 principal components (PCs) were calculated and batch-corrected using Harmony 0.0.5. The PCs served as the basis for k-nearest neighbor calculation (sc.pp.neighbors, n_neighbors = 30), which were used as input for UMAP layout (sc.tl.umap, min_dist = 0.3). Cell populations were clustered using scanpy.tl.leiden, and a suitable resolution was chosen for the main cell type annotation. Cluster marker genes were calculated using a pseudobulk approach, comparing aggregate counts with 2 pseudo-replicates for each cluster to all remaining cells (pyDe-SEQ2 0.4.8). Finally, expression of select marker genes was plotted using Matplotlib 3.8.4 ("imshow"), and clusters were annotated accordingly. The AnnData object was subset to MSCs and ECs, respectively, and the process detailed above for clustering and annotation was repeated, using the top 2000 HVGs and 50 PCs. Clusters were annotated at Leiden resolutions 0.05 and 0.3, for ECs and MSCs, respectively.

Differentially expressed genes were calculated using a pseudobulk approach, comparing aggregate counts with 2 pseudoreplicates for WT and KO each (pyDeSEQ2 0.4.8). DE results were filtered for sex-specific genes due to the presence of cells from mixed-sex samples, and 7 genes were masked accordingly (Gm47283, Xist, Tmsb4x, Ddx3y, Eif2s3y, Uty, Kdm5d). Enrichr (https://maayanlab.cloud/Enrichr/) was used to calculate gene set enrichment via the gseapy 0.10.8 "enrich" API. Up-, ($P < 0.01$, log2FC > 0.5), down- ($P < 0.01$, log2FC < −0.5), and deregulated (up or down) gene sets were tested separately.

## Cell culture

All cell culture experiments were in compliance with S1 regulations. Cultured cells were maintained in a 37 °C incubator with 5% $CO_2$. Human brain vascular smooth muscle cells (HBVSMCs, ScienCell, cat. #1100) were obtained as passage 0 (P0) and stored in liquid nitrogen. Cells were thawed and subcultured according to the manufacturer's instructions.

Cells were expanded up to passage 2 (P2) and frozen at this stage (stock). Freezing media was the usual Smooth Muscle cell Complete Media (SMCM; ScienCell, cat. no. #1101) supplemented with 10% FCS and 10% DMSO. From each T75 plate, 4 stock vials were prepared and frozen. For all experiments, a P2 vial was thawed in a poly-lysine coated T75 plate and expanded with complete media. From P4 to the end of experiments, cells were cultured with smooth muscle cell differentiation media (SMCM supplemented with 2% FCS and 20 ng/mL TGFβ−1). Passage 4 and 5 were used to increase the number of cells and to allow smooth muscle cell contractile differentiation. This was achieved by seeding 15000 cells/cm² in poly-lysine coated T75 dishes and culturing them until reaching confluency with smooth muscle cell differentiation media. Functional experiments were performed at P6-P7.

## Quantitative RT-PCR (qPCR)

Whole mouse embryos were flash frozen in liquid nitrogen, 400 μL of lysis buffer were added to each sample before mechanical dissociation (Ultra-Turrax, IKA T25) and centrifugation ($300 \times g$ for 10 min). The supernatant was collected for RNA isolation. For in vitro samples, cells grown in multi-well 24 cell culture plates were washed twice with sterile PBS, and 200 μL of freshly prepared 1X RNA protection buffer per well were added. Cells were manually disrupted with a pipette tip, collected in 2 mL round-bottom tubes, and stored at − 80 °C until RNA isolation. RNA isolation was performed using the Monarch Total RNA Miniprep Kit (New England Biolabs, cat. #T2010S) according to the manufacturer's instructions.

After RNA isolation, the concentration of RNA was measured (NanoDrop 8000, ThermoFisher). cDNA was synthesized from 1 μg of RNA by reverse transcription (LunaScript RT SuperMix kit, New England Biolabs, cat. #T2010S). For gene expression analysis in HBVSMCs, the following Taqman probes were used: Human *GAPDH-VIC* (ThermoFisher, 4326317E), Human *ELMO2*-FAM (ThermoFisher, Hs00223006_m1), Human *ELMO1*-FAM (ThermoFisher, Hs00404992_m1), Human *ACTA2*-FAM (Thermo-Fisher, Hs00426835_g1), Human *TAGLN*-FAM (ThermoFisher, Hs01038777_g1), Human *NES*-FAM (ThermoFisher, Hs04187831_g1), Human *PDGFRB*-FAM (ThermoFisher, Hs01019589_m1).

Two different pairs of custom-designed primers flanking exon 7 of *Elmo2* were used for SYBR Green-based analysis of gene expression in control and mutant mouse embryos. Primer 1 forward: CTGATG GAAAGGACCCAGTCA; reverse: AACTCCGTGGCGAAAGTCAC. Primer 2 forward: GAGAGTGGGACCAAGCTCCT; reverse: CTCTCTAG

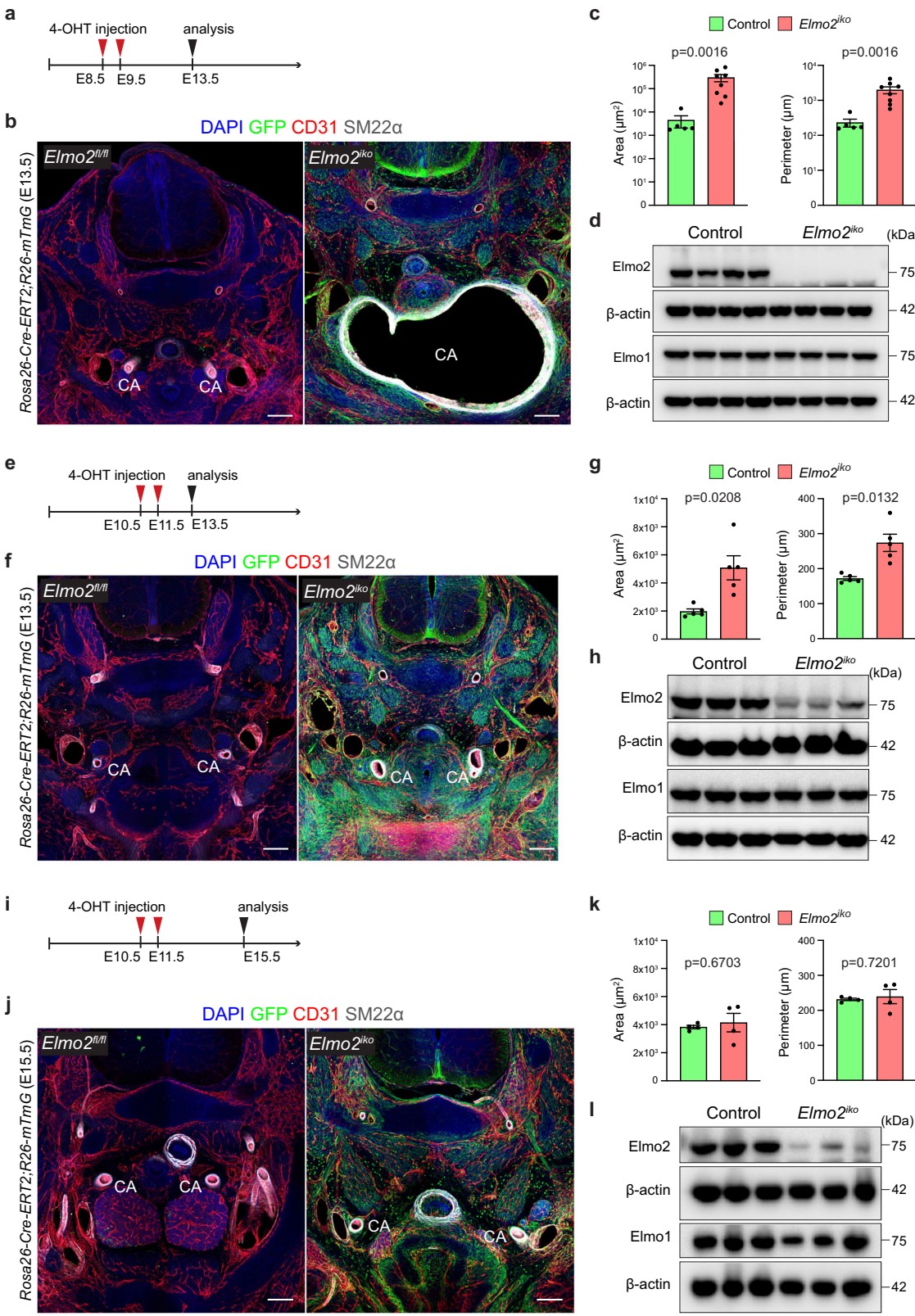

GATGGCCAGGGA. *Gapdh* was used as the housekeeping gene. Forward primer: CCAATGTGTCCGTCGTGGAT; reverse primer: TGCCTGCTTCACCACCTTCT).

## Protein isolation

Whole embryos were flash frozen in liquid nitrogen and stored at −80 °C. Immediately before protein isolation, 400 µL of lysis buffer

(20 mM Tris-HCl; pH 8.0, 150 mM NaCl, 0.5% TritonX-100, 0.1% SDS, 0.1% Na-DOC, 2 mM EDTA, 0.1 mg/mL DNase supplemented with protease and phosphatase inhibitors) were added to frozen samples and subjected to mechanical disruption with Ultra-Turrax (IKA T25) followed by incubation at 4 °C for 30 min on a rotating wheel. Samples were centrifuged at full speed for 15 min to obtain the lysate. The supernatant was collected in a 2 mL tube, and 10 µL of lysate were used

**Fig. 7 | Timing of global *Elmo2* deletion controls aneurysm formation. a** Scheme of 4-hydroxytamoxifen (4-OHT) injection (red arrowheads) and sample collection (analysis, black arrowhead) timepoints. **b** Representative overview confocal images from cross-sections sections of control and *Elmo2^{iko}* E13.5 embryos stained for nuclei (DAPI, blue), recombined cells (GFP, green), endothelial cells (ECs, CD31, red) and vascular smooth muscle cells (VSMCs, SM22α, gray) showing aneurysm formation and merging of carotid arteries (CA) in mutant embryos. Scale bars, 200 μm. **c** Quantitation of area and perimeter of CAs from control and *Elmo2^{iko}* E13.5 embryos after 4-OHT injection at E8.5-E9.5. Mean ± SEM, *n* = 6. Mann-Whitney test. **d** Immunoblot of whole-tissue lysates from control and *Elmo2^{iko}* E13.5 embryos after 4-OHT injection at E8.5-E9.5. Molecular weight marker (kDa) is indicated. **e** Scheme of 4-OHT injection (red arrowheads) and sample collection (analysis, black arrowhead) timepoints. **f** Representative overview confocal images from cross-sections of control and *Elmo2^{iko}* E13.5 embryos stained for nuclei (DAPI, blue), recombined cells (GFP, green), ECs, (CD31, red), and VSMCs (SM22α, gray) showing mild dilation

of carotid arteries (CA) in mutant embryos. Scale bars, 200 μm. **g** Quantitation of area and perimeter of CAs from control and *Elmo2^{iko}* E13.5 embryos after 4-OHT injection at E10.5-E11.5. Mean ± SEM, *n* = 5. Welch's *t*-test. **h** Immunoblot of whole-tissue lysates from control and *Elmo2^{iko}* E13.5 embryos after 4-OHT injection at E10.5-E11.5. Molecular weight marker (kDa) is indicated. **i** Scheme of 4-OHT injection (red arrowheads) and sample collection (analysis, black arrowhead) timepoints. **j** Representative overview confocal images from cross-sections of control and *Elmo2^{iko}* E15.5 embryos stained for nuclei (DAPI, blue), recombined cells (GFP, green), ECs, (CD31, red) and VSMCs (SM22α, gray) showing no obvious phenotypic changes in the carotid arteries (CA) of mutant embryos. Scale bars, 200 μm. **k** Quantitation of area and perimeter of CAs from control and *Elmo2^{iko}* E15.5 embryos after 4-OHT injection at E10.5-E11.5. Mean ± SEM, *n* = 4. Welch's *t*-test. **l** Immunoblot of whole-tissue lysates from control and *Elmo2^{iko}* E15.5 embryos after 4-OHT injection at E10.5-E11.5. Molecular weight marker (kDa) is indicated.

for protein quantitation using ADV02 precision red advanced protein assay reagent (Cytoskeleton, cat. # ADV02). Equal protein concentrations were obtained by diluting the samples with lysis buffer. 3x Laemmli buffer (0.25 M Tris base, 8%SDS, 40% glycerol, 20% β-mercaptoethanol, and 4 mg/mL of bromophenol blue) was added to samples (1/3 volume of sample), mixed well, and boiled at 95 °C for 5 min. After allowing the samples to cool down for 5 min, they were stored at −20 °C until Western blotting.

For in vitro samples, cells were placed on ice and washed twice with ice-cold PBS, followed by the addition of lysis buffer and incubation at 4 °C for 20 min. For a multiwell-6 plate (mw6), 2 mL of PBS per well were used for washing, and 150 μL of lysis buffer per well were used for lysis. Cells were scraped using a precooled cell scraper, and lysates were collected in 1.5 mL tubes. Lysates were sonicated with an amplitude of 80 with 10 pulses per sample, followed by centrifugation at full speed for 10 min at 4 °C. Supernatant was collected in a 1.5 mL tube. Quantitation and storage were performed as described above.

### Western blotting
For Western blotting, upper stacking gel (1.5 M Tris HCL, 0.4% SDS dissolved in water, pH 8.8) and lower resolving gel (0.5 M Tris HCl, 0.4% SDS dissolved in water, pH 6.8) were prepared. 20 kDa to 124 kDa proteins were separated in 10% resolving gels, whereas 125 kDa to 250 kDa proteins were separated in 8% resolving gels. Equal amounts of protein sample were briefly heated at 95 °C, cooled down at RT for 2 min, spined down for 15 sec, and loaded into the wells of the gel. During the electrophoresis run along the stacking gel, the voltage was set to 80 V and to 120 V once the samples reached the resolving gel. After obtaining the required size separation, samples were transferred to an activated PVDF membrane (activated in methanol for 15 sec) over 1.5 h at 25 V. The membrane was blocked with 1% skim milk for 1 h at RT and probed with primary and HRP-conjugated secondary antibodies. ECL prime reagent was used for the detection of bands.

### Gene knockdown
Silencer select *siELMO2* (ThermoFisher, cat. #Hs00223006_m1), which covers most of the *ELMO2* splice variants, was used for knockdown experiments by forward transfection. Cells were seeded at a density of 6000 cells/cm² and cultured overnight. The following day media was exchanged before transfection. A mixture of DMEM-diluted siRNA (24 nM) and lipofectamine RNAiMax (Invitrogen, cat. #13778150) was added drop-wise to the cells, and the plate was placed in the incubator overnight after proper mixing. The following day, the media was refreshed, and cells were kept in culture until the collection time, with media changes every other day.

### Immunofluorescence for cell culture
Cells were washed twice with PBS (without Ca²⁺ and Mg²⁺) and fixed with 2% PFA (filtered through 0.22 μm syringe) at RT for 15 min.

Permeabilization was achieved by incubating cells with 2% Triton X-100 for 15 min at RT. At the end of incubation cells were washed once with PBS and blocked with 1% BSA in PBS for 30 min. Primary and secondary antibody incubations were done at 4 °C overnight.

### Gel contraction assay
Control and *ELMO2* knockdown cells were trypsinized 96 h after siRNA treatment and counted. $5 \times 10^{05}$ cells from each sample were transferred to 1.5 mL tubes placed on ice, and the final volume was adjusted to 500 μL with media. In a 2 mL round-bottom tube placed on ice, 500 μL of collagen-gel forming media (250 μL of 8.69 mg/mL Collagen I, 190 μL of SMC- differentiation media, and 60 μL of 0.1 M NaOH) were prepared. Cells and collagen-gel forming media were mixed in a 1:1 ratio, and 500 μL of this mixture were immediately added to a well of an mw24 plate kept at RT. After 1 h, solidified gels were dislodged from the walls of the wells, and 500 μL of media were added on top. Images of gels at *t* = 0 were acquired with a stereo microscope. After a 12 h incubation at 37 °C, gel contraction was observed and the corresponding images acquired (*t* = 12 h).

For the rescue experiment, 96 h after knockdown, *siControl* and *siELMO2* cells were treated with 100 nM Jasplakinolide (Sigma, cat. #J4580) for 1 h. Subsequently cells were washed twice with PBS and cultured in normal media. After 2 h cells were trypsinized and collected for gel contraction assay as described above.

The area of the gels at *t* = 0 and *t* = 12 h was measured with Fiji, and the percentage of gel contraction was calculated.

### Carbachol-induced contraction
*siControl* and *siELMO2* knockdown cells were labelled 48 h after transfection with 1 nM CellTracker-Green (Invitrogen, cat. #C7025) and CellTracker-Orange (Invitrogen, cat. #C34551), respectively. After a 24h-long incubation, cells were trypsinized and counted. 6000 cells from each sample were mixed and added to a polylysine-coated Ibidi 8-well μ-slide (Ibidi, cat. #80806) and allowed to attach. 96 h after transfection, the media was refreshed and the μ-slide was transferred to a live-imaging microscope equipped with controlled temperature (37 °C) and CO₂ levels (5%). Once the imaging positions and focal plane were defined, media supplemented with 1 mM carbachol was added, and the contraction response of the cells imaged during 15 min. The area of individual cells before carbachol stimulation (*t* = 0) and 15 min after treatment (*t* = 15 min) was measured using Fiji to calculate the percentage of contraction.

### Spreading and attachment assay (2D)
*siControl* and *siELMO2* treated cells were labelled 48 h post-transfection with CellTracker reagents as described before. 96 h after transfection, cells were trypsinized and counted. 6000 cells from each condition were mixed and added to an ibidi 8-well μ-slide coated with 2% collagen. Attached cells were fixed at different time points (10 min,

30 min, 1 h, 2 h, and 6 h) after seeding. For live-imaging experiments, 6000 cells treated with *siControl* or *siELMO2* siRNA were mixed and added to a collagen-coated ibidi 8-well μ-slide, which was imaged overnight under usual cell culture conditions (37 °C with 5% $CO_2$).

For the rescue experiment, 96 h post-transfection, *siControl* and *siELMO2* cells were treated with 100 nM Jasplakinolide (Sigma, cat. #J4580) for 1 h. Subsequently cells were washed twice with PBS and fresh media was provided. After 30 min cells were trypsinized and collected for attachment assay. Cells were seeded into collagen-coated wells and, after 10 min, attached cells were fixed.

### Morphology analysis of cells in fibrin gels (3D)

*siControl* and *siELMO2* treated cells were labelled 48 h post-transfection with CellTracker reagents as described before. Cells were trypsinized 96 h after transfection and counted. 6000 cells from each condition were mixed and resuspended in 1 mL of media. Fibrinogen (Sigma, cat. #F8630) was prepared at a final clottable concentration of 10 mg/mL and sterilized by passing the solution through a 0.22 μm syringe filter before further dilution to the final working concentration (2 mg/mL in PBS). 968 μL of fibrinogen were mixed with 32 μL of aprotinin (Sigma, cat. #A1153, 4 U/mL) and the whole volume (1 mL) mixed with the cell suspension. 250 μL of this mixture were added to each well of an ibidi 8-well μ-slide, which is pre-loaded with 2 μL of thrombin (Sigma, cat. #T4648, 0.1U/μL). Gel polymerizes after a 30 min incubation at RT. 100 μL of media were added on top of the gels, and the plates were incubated overnight at 37 °C and 5% $CO_2$. The next day, the gels were fixed with 2% PFA and imaged.

### Rac1 activation assay

For the analysis of Rac1 activation, the luminescence-based G-LISA Rac-1 activation assay biochem kit (Cytoskeleton Inc., cat. #BK126) was used. Protein samples were collected from *siControl* and *siELMO2*-treated cells and snap frozen in liquid nitrogen until the day of experiment. 10 μL of protein lysate were used for assessing protein concentration using the AVD02 reagent or the reagent included in the kit. Protein sample preparation and G-LISA was performed according to the manufacturer's instructions.

### Analysis of G-actin to F-actin ratio

For G-actin to F-actin ratio analysis cells were lysed in an actin-stabilizing lysis buffer (50 mM PIPES, pH 6.9, 50 mM NaCl, 5 mM $MgCl_2$, 5 mM EGTA, 0.2 mM dithiothreitol, 0.1% NP40, 0.1% Tween 20, 5% glycerol, 1 mM ATP, and protease inhibitors). Lysates were subjected to ultracentrifugation (150,000 × $g$) at 4 °C for 70 min. The supernatant (G-actin fraction) was collected to pre-labelled tubes and stored at 4 °C until protein quantitation. 200 μL of actin depolymerizing buffer (50 mM PIPES, pH 6.9, 5 mM $MgCl_2$, 10 mM $CaCl_2$, 5 μM cytochalasin D) were added to the pellet (F-actin fraction), and this was solubilized by sonication. An equal amount of G-actin and F-actin fractions from control and knockdown cells were loaded to an SDS-PAGE gel, transferred to a PVDF membrane, and stained for β-actin to determine the amount of G-actin and F-actin.

### Quantification and statistical analysis

In all quantifications, the entity of "n" is biological replicates. For in vivo data, each biological replicate corresponds to a single individual (i.e., mouse embryo). In all cases, control and mutant embryos from at least 2 different litters were analyzed. For in vitro data, each biological replicate corresponds to independently treated and processed cell culture samples. Quantitative data is reported as mean ± SEM and is derived from the analysis of multiple samples or technical replicates from each biological replicate. All the data presented in the manuscript without direct quantification was obtained from at least three independent biological replicates.

Statistical analyses were performed with Graphpad Prism10 v.10.2.1 (Perkin Elmer). Analysis of distribution was done using the D'Agostino & Pearson, Anderson-Darling, Shapiro-Wilk or Kolmogorov-Smirnov normality tests, depending on sample size. Based on these results, two-tailed parametric or non-parametric tests were used for the comparison of two groups. For groups with normal distribution and equal variance, the unpaired *t*-test was used, whereas Welch's *t*-test was used for groups with normal distribution and unequal variance. The Mann-Whitney test was used for comparing groups in which at least one did not show a normal distribution. Comparisons among multiple groups were done using either one-way ANOVA or Browne-Forsythe and Welch ANOVA with a specific post-hoc test for multiple comparisons. In the case of repeated measures (time-lapse in vitro), a paired *t*-test was used. Statistical significance was assessed with a 95% confidence interval. Effect sizes were calculated using eta squared for unpaired and Welch's *t*-tests, partial eta squared for paired *t*-test, R squared for ANOVA (both one-way or Browne-Forsythe and Welch), and rank-biserial correlation (r) for Mann-Whitney tests. The source data used for quantitative analysis, as well as detailed statistical results including effect sizes and degrees of freedom, are provided in the Source Data file. In exceptional cases (Fig. 6a and Supplementary Fig. 12a, b, 72 h timepoint), Welch's *t*-test was used for comparison of in vitro samples in which the control group failed to show a normal distribution due to low variance and small sample size ($n = 3$). Distribution was assumed to be normal based on the analysis of the technical replicates of each biological replicate.

Image analysis, processing, and quantitation was performed with Fiji (ImageJ) using basic segmentation tools and morphometry. Evaluation of VSMCs' actin filaments orientation with respect to the ECs' longest axis was done using the Directionality plugin. In the resulting histogram, the bar's height (Amount) represents the frequency of data points corresponding to a given relative directionality angle (°) between the actin filaments and the EC plane. The values shown are normalized based on the assumption that the largest possible angle from the intersection of the planes corresponds to a perpendicular orientation (90°).

### Statistics and reproducibility

All qualitative data presented in this manuscript were obtained from a minimum of three independent biological replicates. Microscopy-based experiments were independently repeated at least three times with consistent results. Western blot experiments were independently performed at least twice, yielding similar outcomes. For all quantitative data, the number of biological replicates (n) is explicitly stated in the corresponding figure legends.

### Reporting summary

Further information on research design is available in the Nature Portfolio Reporting Summary linked to this article.

## Data availability

The single-cell RNA-seq data generated in this study is deposited in the NCBI GEO repository under accession number GSE278960. All measurements used for quantification are provided in the Source Data file along with the results of the different statistical analysis performed. Source data are provided in this paper.

## Code availability

The custom code used for scRNA-seq analysis in this study is based on existing packages and own contributions. It is deposited in a publicly available database and can be accessed through the following link: https://keeper.mpdl.mpg.de/d/c6072badc24b42e5b138/. Dependencies "scrna-tools" and "anndataview" can be found at https://github.com/Bioinformatics-Service-MPI-Munster.

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

## Acknowledgements

We are grateful to the following Service Units of the Max Planck Institute for Molecular Biomedicine (MPI-BM, Münster, Germany): Animal Facility, Flow Cytometry, BioOptic Service, Sequencing, and Bioinformatics. We thank Dr. Stefan Volkery and Dr. Nils Kirschnick (MPI-BM) for their technical support during light sheet microscopy and Anja Michelbach from the group of Prof. Dr. Sara Wickström (MPI-BM) for the G-actin:F-actin assay protocol. The study was supported by the Max Planck Society (R.H.A.), the European Research Council (AdG 101139772, PROTECT; R.H.A.), the DFG (CRC 1366, project no. 394046768; R.H.A.), and the Cells in Motion (CiM) graduate school (A.S.).

## Author contributions

A.S., R.D.H., and R.H.A. designed the study. A.S. performed the majority of the experiments. H.A. and R.D.H. performed some of the experiments. K.K. performed the bioinformatic transcriptomic analysis and wrote the corresponding methods. R.D.H. supervised A.S. during the study. A.S., R.D.H., and R.H.A. wrote the manuscript.

## Funding

## Competing interests

The authors declare no competing interests.
