## [Peer Review file · Nature Communications]

ELMO2 is an essential regulator of carotid artery development

Corresponding Author: Professor Ralf Adams

Version 0:

Reviewer comments:

Reviewer #1

(Remarks to the Author)

The paper by Suresh et al. examines the role of ELMO2, a cytoskeletal adaptor protein, in vascular development and in vascular smooth muscle cells (VSMC). This is an elegant and well-done study whose initial results were confusing, leading to further elegant work to dissect a complex but biomedically important process. The work describes an elegant series of genetic experiments that tackle the complexity of great artery development, and specifically the development of the carotid arteries from the 3rd pharyngeal arch artery (PAA). The starting point is a link between a hereditary human disease, intraosseous vascular malformation (VMOS) and ELMO2, and knowledge that the global deletion is mid-gestation embryonic lethal. A careful analysis of the global deletion in mice reveals specific effects on carotid artery diameter and aberrant SMC and EC properties. However, the phenotype is only recapitulated with neural crest deletion of ELMO2 (not EC or VSMC deletion), and cell data reveals mis-regulation of adhesion and contractility in VSMC. Interesting potential reasons for the specificity are postulated in the discussion. This work is quite significant as it shows how developmental defects may lead to post-birth pathology in novel ways, and it suggests that primary vascular defects may affect bone development, as the human disease is associated with overgrowth of jaw bones. The work uses state-of-the-art genetic, cell-based and bioinformatic tools that largely support the conclusions and claims. Great artery development is complex and involves multiple sources of embryonic tissue, and temporal aspects are highly regulated, making the genetic experiments non-trivial to design and interpret. The methodology is sound and with enough detail to reproduce the work. A few suggestions would strengthen this novel and impactful study:

Minor:

1. It's not clear whether the specificity is thought to be from the embryonic origin (NCC) or only the timing (relatively early in PAA development), perhaps a little more discussion of these points.
2. Figure 2 could be better labeled for embryonic stage.
3. If the mutant (or siRNA silenced) VSMC do not contract properly, is it surprising that there is not a transcriptional change in the "contractile" phenotype?

Reviewer #2

(Remarks to the Author)

Suresh et al. present an insightful study on the role of the cytoskeletal adaptor protein ELMO2 in carotid artery formation. Early embryonic global or neural crest cell-specific inactivation of Elmo2 in mice disrupts the formation of the third pharyngeal arch arteries (PAA). Interestingly, however, deletion in ECs or differentiated SMCs does not replicate this phenotype in vivo. From careful analysis of the phenotype, the authors conclude that ELMO2 is not required for NCC migration into the pharyngeal arches, vascular SMC specification, identity, or survival. Instead, it specifically controls the contractile and stabilizing functions of NCC-derived SMCs forming around PAAs – a finding that is further substantiated through experiments in cultured SMCs in vitro. These findings provide important insight into understanding pathogenesis of intraosseous vascular malformation (VMOS) linked to ELMO2 loss-of-function mutations. The manuscript is convincingly presented, well-written, and easy to follow. The data are robust and carefully interpreted, avoiding overstatements.

Major comments:

1. Apart from SMCs, NCCs give rise to most connective tissues in the head and neck region. If feasible, performing mosaic analysis using for example iSuRe-Cre (or the recently published iSuRe HadCre) technology to delete Elmo2 would further validate the identity and role of NCC-derived cells linked to the ELMO2 loss-of-function phenotype.
2. It appears as if lymph sacs might be blood-filled in the Elmo2 mutant embryos (Fig. 1a). This could be secondary to the other vascular defects, but if this is the case please note in the text.
3. Co-staining for a cell membrane marker would be helpful to appreciate abluminal mislocalization of Podocalyxin in Fig. 2d.

Minor comments:

1. Please provide details for super-resolution confocal microscopy (line 160) in methods.
2. Endomucin has been reported to be expressed in the aorta during early developmental stages (PMID: 11747076), staining in the dilated carotid arteries of Elmo2^{-/-} embryos (Fig. 1e and 2a) may thus reflect failure in downregulation.
3. Consider adding 'Elmo2 expression' in Fig. 3c for clarification.

Reviewer #3

(Remarks to the Author)

In this manuscript, the authors investigate the role of ELMO2 during embryonic development in mice. Global deletion of Elmo2 leads to embryonic lethality after midgestation which the authors attribute to severe dilation of the 3rd pharyngeal arch arteries and aneurysm formation in the common carotid arteries, the only vessels that appear to be affected.

Using a floxed allele of Elmo2 in combination with various Cre lines, the authors try to pin-point the cell type responsible for the phenotype as well as the period during which Elmo2 is indispensable for normal development and arrive at the conclusion that Elmo2 is most likely required in vascular smooth muscle cells during a short time window in early embryonic development. While vascular smooth muscle cell abundance and differentiation does not seem to be affected vascular smooth muscle cell organization appears abnormal in the affected vessels.

The experiments are carefully executed and the imaging is of very high quality. The phenotype affecting the 3rd pharyngeal arch arteries and the common carotids are well described. However, the causal relationship between the rather mild vascular smooth muscle cell defects and the severe vascular anomalies is less convincing and direct evidence that Elmo2 is specifically required in vascular smooth muscle cells *in vivo* is, unfortunately, missing.

Major points:

1. While ELMO2 silencing in human vascular smooth muscle cells leads to impaired contractility and defects in actin remodeling *in vitro*, the corresponding *in vivo* evidence is missing. Expression of contractile genes in vascular smooth muscle cells seems not to be affected in the single-cell data and immunohistochemistry for markers of contractility appears unchanged *in vivo*.
2. Deletion of Elmo2 specifically in vascular smooth muscle cells using the Tagln-Cre line does not recapitulate the global knockout. While the timing of expression in the Tagln-Cre line may indeed be too late to generate a phenotype, could the authors use their PDGFR-betaCreERT2 line to obtain an earlier deletion in vascular smooth muscle cells and pericytes and provide direct evidence for their involvement in the phenotype?

In the absence of more direct evidence that the phenotype is caused by Elmo2 function specifically in vascular smooth muscle cells, I recommend exercising caution before proposing a causal relationship.

3. The authors seem to conclude from their EdU labeling experiments that the increase in EC and vSMC proliferation that they observe is a mere reflection of vessel dilation? (since it's no longer significant after normalizing to vessel perimeter line 170-178). Can this conclusion really be drawn? Under the assumption that the endothelial cells proliferate at the same rate per perimeter unit and that no other factors are influencing the growth, the larger blood vessel will have a higher total number of proliferating endothelial cells, and therefore, it will grow faster in terms of total endothelial mass. The massively enlarged 3rd pharyngeal arch arteries and carotid arteries must contain substantially more EC and vSMCs than the corresponding vessels in control mice and these cells must come from somewhere, so at one point proliferation rates had to be increased to create more cells to start with. Could it be that the authors missed the period at which EC and vSMCs proliferated at higher rates in Elmo2 mutant mice? Could an earlier analysis of proliferation for example at E11.5 be worth investigating? Does the single-cell RNA seq. data provide any further insight into proliferation of EC and vSMCs?

Minor points:

1. In order to better understand from which regions of the embryos the tissue sections have been taken, a schematic drawing or alternatively a line in the embryo whole mount picture would be helpful in Figure 1.

2. A schematic representation (or table) summarizing which cell types are targeted by the different Cre lines and a summary of phenotypes (including different time points if applicable) would further improve the paper.
3. In Fig 1G, the Prox-1 positive jugular lymph sac seems enlarged and hyperplastic in Elmo2 mutants. Are these vessels entirely normal?
4. Along those lines, several other vessels (not only the carotid arter) appear enlarged in Supplementary Figure 6b.
5. It would be appreciated if the time-lapse movies were merged and the speed was reduced to better follow the differences between control and Elmo2 knockdown cells. The movies also appear to suffer from focus drift, which makes it difficult to appreciate the phenotypical differences between control and Elmo2 knockdown cells.
6. In Supplementary Figure 1f the labeling of the carotid arteries is missing. It would be helpful to the reader if the labeling could be added, for consistency.
7. In Line 144/Figure 2e and f: Ectopic expression of SM22a in ECs look not fully convincing to me. It rather looks like that vascular smooth muscle cells invade the endothelial layer rather than ECs start to express vSMC specific genes. Maybe a zoom in panel and another color combination e.g. green/violet could help to clarify this. Do these cells also express other vSMC markers?
8. TH-IRES-Cre mediated Elmo2 deletion is described but not shown (only the lineage tracing is shown. Supplementary Figure 10d). Could the authors add this data?
9. The authors use phospho-myosin light chain 2 (phMLC2) as a functional marker for vSMC contraction. Could the authors please add an appropriate reference for that? Based on available single-cell datasets expression of Myl2 in vSMCs is rather weak. Maybe phMYL9 would be a better marker?

Reviewer #4

(Remarks to the Author)

Version 1:

Reviewer comments:

Reviewer #2

(Remarks to the Author)

The authors have carefully addressed all reviewers' comments and I have no further concerns. Congratulations on an insightful and well-executed study.

Reviewer #3

(Remarks to the Author)

The authors have addressed all my questions and concerns. I highly appreciate the effort that the authors put into revising their manuscript.

Reviewer #4

(Remarks to the Author)

RESPONSE TO REVIEWERS

First of all, we would like to show our appreciation and gratitude to the appointed reviewers for their time and thoughtful suggestions, which have been instrumental for improving our manuscript. While we provide below a detailed point-by-point response to each comment, we would like to summarize the most important changes and additions that are present in the revised version of the paper.

Changes in the **Results** section:

- a) Brief mention to the pan-endothelial expression of Endomucin during early development in connection with the persistence of this marker in arterial territories of *Elmo2*-mutant embryos.
- b) Additional analysis of proliferation at E11.5 and revised interpretation of these results.

Changes in the **Discussion** section:

- a) The potential importance of the embryonic origin of mural cells in relation with the specific nature of the vascular phenotype is discussed.
- b) Revised phrasing to avoid overstating the relationship between *Elmo2* inactivation and VSMC contractility *in vivo*.

Changes in the **Methods** section:

- a) Inclusion of details for super-resolution microscopy and specifics about quantitation approaches.

Changes in the **Main Figures**:

Fig. 1f: Schematic representation of embryos at the different stages analyzed indicating the area imaged and its orientation.

Fig. 2: Improved layout to ease following the panels' order as they are described in the text.

Fig. 2a: Indication of the embryonic stage in the different panels.

Fig. 2d: New panel showing unambiguous expression of SM22 α in Sox17⁺ arterial ECs.

Fig. 3c: Inclusion of header "*Elmo2* expression" to ease data interpretation.

Changes in the **Supplementary Figures**:

Supplementary Fig. 1f: Labels have been added to indicate carotid arteries.

Supplementary Fig. 1g: New panel showing secondary effects of the *Elmo2*-global knockout phenotype on jugular vein and jugular lymph sac.

Supplementary Fig. 3: Additional panels showing unaffected expression of differentiation markers in VSMCs and normal phosphorylation of myosin light chain 9.

Supplementary Fig. 4d-f: New panels showing the analysis of EC and VSMC proliferation in E11.5 control and *Elmo2*-knockout embryos.

Supplementary Fig. 6: New data from vibratome sections replacing the former panels, which were obtained from cryosections.

Supplementary Fig. 11: Shows the phenotypic characterization and lineage-tracing analysis of *TH-IRES-Cre*-mediated recombination. The supplementary *in vitro* data information is now shown in Supplementary Fig. 12 of the revised manuscript.

Supplementary Fig. 12: Additional panel showing unaffected phosphorylation of myosin light chain 9 *in vitro*.

Time-lapse videos have been repeated in order to increase spatial and temporal resolution. The new file includes both control and knockdown cells side by side in a merged video that replaces the old version.

Addition of Supplementary Table 1. Summary of *in vivo* phenotypes upon *Elmo2* deletion, which provides an overview of the phenotypic outcome of the different genetic mouse models employed along the manuscript.

Moreover, we have taken the opportunity to further increase the robustness of some quantitations by including additional samples generated during the revision experiments. Likewise, statistical approaches and stated p-values have been carefully revised during preparation of the source data file and figure legends have been edited to meet the required criteria.

We are convinced that the new experiments performed, together with the detailed revision of the text and improvement of the figures, thoroughly addressed all critical concerns raised and allowed us to further improve the quality and relevance of the present study.

In the following point-by-point response, the reviewers' comments are copied in full and each comment is followed by our reply (blue text).

Reviewer's Comments

Reviewer #1 (Remarks to the Author):

The paper by Suresh et al. examines the role of ELMO2, a cytoskeletal adaptor protein, in vascular development and in vascular smooth muscle cells (VSMC). This is an elegant and well-done study whose initial results were confusing, leading to further elegant work to dissect a complex but biomedically important process. The work describes an elegant series of genetic experiments that tackle the complexity of great artery development, and specifically the development of the carotid arteries from the 3rd pharyngeal arch artery (PAA). The starting point is a link between a hereditary human disease, intraosseous vascular malformation (VMOS) and ELMO2, and knowledge that the global deletion is mid-gestation embryonic lethal. A careful analysis of the global deletion in mice reveals specific effects on carotid artery diameter and aberrant SMC and EC properties. However, the phenotype is only recapitulated with neural crest deletion of ELMO2 (not EC or VSMC deletion), and cell data reveals misregulation of adhesion and contractility in VSMC. Interesting potential reasons for the specificity are postulated in the discussion. This work is quite significant as it shows how developmental defects may lead to post-birth pathology in novel ways, and it suggests that primary vascular defects may affect bone development, as the human disease is associated with overgrowth of jaw bones. The work uses state-of-the-art genetic, cell-based and bioinformatic tools that largely support the conclusions and claims. Great artery development is complex and involves multiple sources of embryonic tissue, and temporal aspects are highly regulated, making the genetic experiments non-trivial to design and interpret. The methodology is sound and with enough detail to reproduce the work. A few suggestions would strengthen this novel and impactful study:

We are grateful for the careful evaluation of our manuscript and appreciate the reviewer's assessment of our study.

Minor:

1. It's not clear whether the specificity is thought to be from the embryonic origin (NCC) or only the timing (relatively early in PAA development), perhaps a little more discussion of these points.

Thank you for this comment. The experiments using tamoxifen-inducible global deletion of *Elmo2* with the *Rosa26-CreERT2* mouse line establish that the timing of gene inactivation is critical for the development of the phenotype. Early inactivation of *Elmo2* from E8.5 to E13.5 reproduces the global KO phenotype (Fig. 7a-d), whereas no alterations were observed when the same experiment is done from E10.5 to E15.5 (Fig. 7i-l). Moreover, the differences in the phenotypic outcome after *Wnt1-Cre2* or *Tagln-Cre*-mediated deletion also supports that the timing of gene inactivation is critical (see comparison of the two Cre lines in Supplementary Fig. 9).

The specificity of the phenotype towards the 3rd PAA is an interesting aspect that cannot be explained exclusively by the ontogeny of its vascular components, as this is common to other PAAs that appear unaffected. Reasons for this difference could be the combined result of its embryonic origin, the high hemodynamic forces that act on the 3rd PAA, and the persistence of this structure through its remodeling into the common carotid arteries. As suggested by the reviewer, we have expanded the discussion to highlight how flow and other features of the 3rd PAA might contribute to the mutant phenotype.

2. Figure 2 could be better labeled for embryonic stage.

We agree and have now indicate the embryonic stages in all image panels in Fig. 2 but also in other figures.

3. If the mutant (or siRNA silenced) VSMC do not contract properly, is it surprising that there is not a transcriptional change in the "contractile" phenotype?

Indeed, it was also surprising to us that the defects in the organization and cellular alignment of *Elmo2* null VSMCs observed *in vivo* did not correlate with profound changes in the expression of markers related to the contractile phenotype. This was also the case for our *in vitro* model in which efficient *Elmo2* silencing in HBVSMCs did not alter the expression levels of typical contractile markers but had a strong effect on the functional capacity of these cells.

Our interpretation of this apparent paradox is that ELMO2 plays an important role in bringing together different molecules that enable cytoskeleton remodeling through actin polymerization. The increase in the proportion of filamentous actin (F-actin) that occurs in response to the stimulation of smooth muscle cells has an essential role in the generation of mechanical tension¹ and the transmission of the forces generated². In this regard, it should be noted that several vasoconstrictors are known to induce an increase in actin polymerization³, whereas pharmacologic depolymerization of F-actin causes VSMC relaxation⁴. Furthermore, defects in Rac1 activation have been shown to reduce the contraction in VSMCs presumably due an inhibition in intracellular calcium oscillations⁵.

Other studies have proposed that functional contractile output is dependent on cellular architecture and that it has no positive correlation with contractile marker expression⁶. Similarly, comparative analysis of vascular smooth muscle cell lines from healthy donors and patients affected by abdominal aortic aneurysms revealed that marker expression is not indicative of phenotypic defects linked to VSMC contraction⁷.

Altogether, these studies show that contractile ability can be impaired without parallel changes in the expression of contractility-associated markers. This is an important aspect reflecting

that transcriptomic assessment of the VSMC phenotype may not always be sufficient for predicting functional behavior.

Reviewer #2 (Remarks to the Author):

Suresh et al. present an insightful study on the role of the cytoskeletal adaptor protein ELMO2 in carotid artery formation. Early embryonic global or neural crest cell-specific inactivation of *Elmo2* in mice disrupts the formation of the third pharyngeal arch arteries (PAA). Interestingly, however, deletion in ECs or differentiated SMCs does not replicate this phenotype in vivo. From careful analysis of the phenotype, the authors conclude that ELMO2 is not required for NCC migration into the pharyngeal arches, vascular SMC specification, identity, or survival. Instead, it specifically controls the contractile and stabilizing functions of NCC-derived SMCs forming around PAAs – a finding that is further substantiated through experiments in cultured SMCs in vitro. These findings provide important insight into understanding pathogenesis of intraosseous vascular malformation (VMOS) linked to ELMO2 loss-of-function mutations. The manuscript is convincingly presented, well-written, and easy to follow. The data are robust and carefully interpreted, avoiding overstatements.

We thank the reviewer for the supportive assessment of our work.

Major comments:

1. Apart from SMCs, NCCs give rise to most connective tissues in the head and neck region. If feasible, performing mosaic analysis using for example *iSuRe-Cre* (or the recently published *iSuRe HadCre*) technology to delete *Elmo2* would further validate the identity and role of NCC-derived cells linked to the ELMO2 loss-of-function phenotype.

We greatly appreciate this suggestion, but it should be noted that the *iSuRe-Cre* and *iSuRe-HadCre* lines have been primarily designed for tamoxifen-inducible genetic experiments. The recently published description of *iSuRe-HadCre*⁸, for example, reports that the line “converts inducible CreERT2 activity into constitutive Cre expression”. Constitutive Cre lines, which have been used for the large majority of our experiments, would lead to very strong/robust Cre activity with a higher risk of toxic side effects⁹.

Moreover, clonal/mosaic analysis is only feasible with inducible CreERT2 (by limiting tamoxifen/4-OHT administration) or other recombinases (such as variants of FLP) with weak and therefore patchy activity. In our case, mosaic analysis would require the inclusion of tissue-specific CreERT2 lines together with *iSuRe-HadCre* or suitable Cre reporters all in the *Elmo2* conditional background. Mosaic analysis, while valuable, falls outside the scope of our current study as it extends beyond our primary research objectives and available resources.

In the revised version, we were able to provide additional data regarding *Elmo2* deletion using *TH-IRES-Cre*, a line that drives recombination in neural crest-derived neuronal populations. Normal phenotypic development of these mutants rules out that the phenotype of global or *Wnt1-Cre2*-induced *Elmo2* mutants is caused by defects in neural crest-derived neuron populations.

2. It appears as if lymph sacs might be blood-filled in the *Elmo2* mutant embryos (Fig. 1a). This could be secondary to the other vascular defects, but if this is the case please note in the text.

Thank you very much for this comment. Indeed, in some instances jugular lymph sacs are filled with blood and the jugular vein appears collapsed due to compression from the massively

enlarged carotid arteries. These secondary effects of the phenotype are now highlighted in Supplementary Fig. 1g and mentioned in the text.

3. Co-staining for a cell membrane marker would be helpful to appreciate abluminal mislocalization of Podocalyxin in Fig. 2d.

Thank you very much for this excellent suggestion, which we have tried to implement. Our search for specific membrane markers of the endothelium led to junctional proteins such as PECAM1 (CD31) and VE-Cadherin. From these, we selected the latter because the CD31 and Podocalyxin antibodies in our study are raised in the same species. Unfortunately, the localization of VE-Cadherin is restricted to contact points between neighboring endothelial cells (ECs) and does not outline the full plasma membrane, precluding a better visualization of the cellular contour (see Revision Fig. 1a below). We therefore opted to use Cre-inducible membrane-anchored GFP for surface labelling of all recombined cells in global *Elmo2* knockout embryos generated with the *PGK-Cre* line. Staining with Podocalyxin allowed identification of ECs and clear visualization of the polarity defects in the mutant embryos (Revision Fig. 1b). Careful comparison of the newly generated data with that presented in Fig. 2d indicates that the panel in the manuscript is representative and already very clear. We trust that the reviewer shares our opinion on this matter.

Revision Fig. 1. Analysis of endothelial cell (EC) polarity upon *Elmo2* global deletion

- a) E12.5 3rd PAA from control and *Elmo2*^{-/-} embryos stained for nuclei (DAPI, blue), EC-junctions (VE-Cadherin, green), VSMCs (αSMA, red) and podocalyxin (PODXL, gray). White arrowheads indicate abluminal PODXL localization in *Elmo2*^{-/-} ECs. Scale bars, 10 μm.
- b) E13.5 CAs from control (*Elmo2*^{+p;PGK-Cre^{+T}) and *Elmo2*^Δ (*Elmo2*^{p/p; PGK-Cre^{+T}) embryos stained for nuclei (DAPI, blue) and podocalyxin (PODXL, magenta). Membrane-tagged GFP expression (green) labels all recombined cells. White arrowheads point to ECs with abluminal PODXL localization in knockout embryos. Scale bars, 5 μm.}}

Minor comments:

1. Please provide details for super-resolution confocal microscopy (line 160) in methods.

Thanks for alerting us to this omission. We have now included the necessary details and also explain the quantitation method employed for analyzing the orientation of VSMCs with respect to ECs based on the super-resolution images.

2. Endomucin has been reported to be expressed in the aorta during early developmental stages (PMID: 11747076), staining in the dilated carotid arteries of Elmo2^{-/-} embryos (Fig. 1e and 2a) may thus reflect failure in downregulation.

We agree with this comment. Actually, this is also our interpretation regarding the persistent expression of Endomucin in arterial territories. We have included in the text a brief mention to the pan-endothelial expression of Endomucin during early stages of embryonic development with the suggested reference.

3. Consider adding 'Elmo2 expression' in Fig. 3c for clarification.

Thank you. We have included this suggestion in the revised version of the manuscript.

Reviewer #3 (Remarks to the Author):

In this manuscript, the authors investigate the role of ELMO2 during embryonic development in mice. Global deletion of Elmo2 leads to embryonic lethality after midgestation which the authors attribute to severe dilation of the 3rd pharyngeal arch arteries and aneurysm formation in the common carotid arteries, the only vessels that appear to be affected.

Using a floxed allele of Elmo2 in combination with various Cre lines, the authors try to pinpoint the cell type responsible for the phenotype as well as the period during which Elmo2 is indispensable for normal development and arrive at the conclusion that Elmo2 is most likely required in vascular smooth muscle cells during a short time window in early embryonic development. While vascular smooth muscle cell abundance and differentiation does not seem to be affected vascular smooth muscle cell organization appears abnormal in the affected vessels.

The experiments are carefully executed and the imaging is of very high quality. The phenotype affecting the 3rd pharyngeal arch arteries and the common carotids are well described. However, the causal relationship between the rather mild vascular smooth muscle cell defects and the severe vascular anomalies is less convincing and direct evidence that Elmo2 is specifically required in vascular smooth muscle cells in vivo is, unfortunately, missing.

We are grateful for the thoughtful and constructive comments. In addition to our answers to the various individual points raised by the reviewer, we would like to start by outlining why developing and therefore immature large arteries might be particularly vulnerable to VSMC defects:

It is well known that the mechanical properties necessary for proper function of large arteries rely on the elastic fiber network organized by smooth muscle cells in the *tunica media*¹⁰. In fact, the formation of aneurysms is characterized by a disrupted vessel wall structure with degraded elastic laminae and disappearance of organized VSMC layers¹¹. Extracellular matrix (ECM) assembly is also a major factor controlling the stability of atherosclerotic plaques¹². In the developing mouse aorta, the deposition of collagen fibers and elastin in the extracellular matrix (ECM) of the arterial wall begins only around E14 and is not established before late

gestation¹³. This observation coincides with our analysis of the ECM composition around the carotid arteries (Revision Fig. 2), reflecting that the basic organization of elastic lamellae is only achieved around E15.5.

Revision Fig. 2. Analysis of the ECM organization in the carotid arteries of wildtype embryos

Comparative analysis of cross-sections from carotid arteries of E15.5 and E13.5 embryos stained for nuclei (DAPI, blue), VSMCs (SM22 α , green), collagen IV (COL IV, red) and elastin (gray). Scale bars, 10 μ m.

These results highlight that the changes in VSMC behavior induced by *Elmo2* inactivation are triggered at a developmental stage in which carotid arteries are not yet supported by organized layers of ECM. Due to the absence of concentric matrix layers and because of the rapid reorganization of the 3rd PAA during this phase of development, the contractile capacity of VMSCs is likely to play a much more critical role than in mature arteries. Moreover, the disorganized arrangement of α SMA⁺ fibers in *Elmo2* knockout embryos is a direct indication of compromised VSMC contractile ability (Fig. 2h), given the known importance of proper cytoskeletal organization for force generation^{1,14}.

Major points:

1. While ELMO2 silencing in human vascular smooth muscle cells leads to impaired contractility and defects in actin remodeling *in vitro*, the corresponding *in vivo* evidence is missing. Expression of contractile genes in vascular smooth muscle cells seems not to be affected in the single-cell data and immunohistochemistry for markers of contractility appears unchanged *in vivo*.

Thank you for this assessment. As mentioned above, we show that in *Elmo2* knockout carotid arteries, the normal perpendicular orientation of VSMCs relative to the underlying endothelium, as well as the parallel alignment of α SMA⁺ fibers (Fig. 2h-j), are disrupted. These structural features are physiologic responses of VSMCs to cyclic stretch and play a fundamental role in the regulation of vascular lumen caliber^{15,16}.

Functional assays, like the analysis of gel contraction *in vitro* (Fig. 6j, k), or contractility measurements of dissected vessels are obviously not feasible for the embryonic carotid artery.

However, while the expression of markers associated with cell contractility is indeed not changed at the transcript and protein level, the normal spatial distribution of α SMA and SM22 α is substantially compromised, as we show in global and NCC-specific *Elmo2* mutants (Fig. 2k, l; Fig. 5e).

Furthermore, *Tagln-Cre*-mediated *Elmo2* inactivation results in dilation of the carotid arteries pointing directly at a VSMC-specific role of the gene product (Fig. 4b-d). We appreciate that this phenotype is much milder than the defects seen in global and NCC-specific *Elmo2* mutants, which is likely to reflect differences in the timing of gene inactivation during embryonic development (Supplementary Fig. 9).

Finally, it should also be noted that the *in vitro* contractility of HBVSMCs upon *Elmo2* downregulation is severely impaired, despite the lack of detectable changes in the expression of relevant components of the contractile machinery. As already pointed out in our response to Reviewer # 1, the remodeling of the actin cytoskeleton and the assembly of F-actin are of crucial importance for transmission of the contractile forces. We have provided evidence of defective actin organization upon *Elmo2* inactivation both *in vivo* and *in vitro*. Moreover, our *in vitro* rescue experiments through pharmacologically-induced stabilization of F-actin further prove the indispensability of proper actin remodeling for VSMC contractility (Fig. 6q and r).

2. Deletion of *Elmo2* specifically in vascular smooth muscle cells using the *Tagln-Cre* line does not recapitulate the global knockout. While the timing of expression in the *Tagln-Cre* line may indeed be too late to generate a phenotype, could the authors use their *PDGFR-betaCreERT2* line to obtain an earlier deletion in vascular smooth muscle cells and pericytes and provide direct evidence for their involvement in the phenotype?

In the absence of more direct evidence that the phenotype is caused by *Elmo2* function specifically in vascular smooth muscle cells, I recommend exercising caution before proposing a causal relationship.

This is a valid and reasonable suggestion that could further demonstrate the specific importance of *Elmo2* in early mural cell progenitors giving rise to VSMCs in the 3rd PAA and carotid arteries. Indeed, these experiments were performed soon after the initial characterization of vascular phenotype but were not included in the original manuscript given certain limitations disclosed below.

In order to assess the specificity and efficiency of the recombination events driven by the *Pdgfrb(BAC)-CreERT2* line in the early embryo, we bred this transgenic line into the *R26-mTmG* background and analyzed the abundance of recombined (GFP⁺) cells within the medial layer of the developing carotid arteries at E13.5 after 4-hydroxytamoxifen (4-OHT) administration from E8.5 (using the same regime as that described for the *R26-CreERT2* mouse model). These results clearly showed that the line was not able to efficiently target the population of interest, with only a few recombined cells being present within the SM22 α ⁺ layer of VSMCs (Revision Fig. 3c). Unfortunately, these limitations in recombination efficiency were not overcome by increased or earlier 4-OHT administration. In addition, it is well established that very high concentrations of tamoxifen are not compatible with maintaining viable pregnancies.

Revision Fig. 3. Limited recombination efficiency in embryonic carotid artery-mural cells using the inducible *Pdgfrb-CreERT2* line

- Overview confocal image of the cervical region from an E13.5 embryo carrying the Cre-reporter allele *R26-mTmG* and induced with 4-OHT at E8.5 and E9.5. Section stained for nuclei (DAPI, blue), recombined cells (GFP, green), endothelial cells (CD31, red) and vascular smooth muscle cells (SM22 α , grey). Carotid arteries (CA) are indicated with a dashed-line box. Scale bars, 200 μ m.
- Maximum intensity projections of high magnification confocal pictures from E13.5 CAs, highlighted in panel (a) showing limited contribution of recombined cells (GFP $^+$) to vascular smooth muscle cells (SM22 α^+) forming the medial layer. Scale bars, 25 μ m.
- Single optical section from the image shown in (b) highlighting co-localization of GFP and SM22 α in the few recombined cells of the medial layer. Scale bars, 25 μ m.

Consistent with the patchy recombination in carotid artery VSMCs shown above, *Pdgfrb(BAC)-CreERT2*-mediated inactivation of *Elmo2*, induced by 4-hydroxytamoxifen (4-OHT) administration from E8.5, did not induce macroscopic changes in the resulting mutants at E13.5 (see Revision Fig. 4a further below). Additional histologic characterization and morphometric analysis confirmed that the size of carotid arteries remained unchanged (Revision Fig. 4b and c) and that the organization of the VSMC layer was not affected (Revision Fig. 4d) in mutant embryos relative to littermate controls.

Taken together, technical limitations of the available Cre lines and the absence of suitable tools allowing efficient targeting of NCC subpopulations and VSMC progenitors *in vivo* preclude additional meaningful genetic experiments at this point.

In the revised Discussion, we also have toned down the statement regarding *Elmo2* function in contractile and vessel-stabilizing NCC-derived smooth muscle cells surrounding the 3rd PAA.

The Discussion also mentions the previously published NCC-specific deletion of *Rac1*¹⁷, which results in aberrant patterning of pharyngeal arch arteries, defective outflow tract septation and severe aneurysms in the vessels branching from the common arterial trunk. Strikingly, this article states that “the aberrant vessels in mutants were also surrounded by wall containing smooth muscle alpha-actin (SM α -A)-positive cells”. Similar to *Elmo2* mutants, loss of *Rac1* impaired cell spreading and cytoskeletal organization *in vitro*. Taken together, our own analysis and published literature argue that the loss of certain genes can impair the function of VSMCs without disrupting the expression of key markers in the highly dynamic setting of embryonic development.

Revision Fig. 4. Deletion of *Elmo2* in mural cells using *Pdgfrb-CreERT2* does not lead to phenotypic alterations in mutant embryos

- Representative images of control (*Elmo2*^{p/p}; *Pdgfrb-CreERT2*^{+/+}) and *Elmo2*^{ΔIMC} (*Elmo2*^{p/p}; *Pdgfrb-CreERT2*^{T/T}) E13.5 embryos showing absence of macroscopic defects in mutant mice after 4-OHT administration from E8.5 to E9.5. Scale bars, 2mm.
- Confocal overview images of E13.5 control and *Elmo2*^{ΔIMC} embryos reveal comparable development and conserved carotid artery (CA) size as well as normal development of jugular veins (JV) and jugular lymph sacs (JLS). Sections stained for nuclei (DAPI, blue), endothelial cells (CD31, green; EMCN, magenta), and vascular smooth muscle cells (αSMA, red; SM22α, grey). Scale bars, 200μm.
- Quantitation of carotid arteries' area and perimeter from E13.5 control and *Elmo2*^{ΔIMC} embryos. Mean ±SEM, n=6 (control), n=11 (*Elmo2*^{ΔIMC}). Unpaired t-test (area) and Mann-Whitney test (perimeter).
- High magnification images of the CAs from E13.5 control and *Elmo2*^{ΔIMC} embryos as stained in (b). Scale bars, 20μm.

3. The authors seem to conclude from their EdU labeling experiments that the increase in EC and vSMC proliferation that they observe is a mere reflection of vessel dilation? (since it's no longer significant after normalizing to vessel perimeter line 170-178). Can this conclusion really be drawn? Under the assumption that the endothelial cells proliferate at the same rate per perimeter unit and that no other factors are influencing the growth, the larger blood vessel will have a higher total number of proliferating endothelial cells, and therefore, it will grow faster in terms of total endothelial mass. The massively enlarged 3rd pharyngeal arch arteries and carotid arteries must contain substantially more EC and vSMCs than the corresponding vessels in control mice and these cells must come from somewhere, so at one point

proliferation rates had to be increased to create more cells to start with. Could it be that the authors missed the period at which EC and vSMCs proliferated at higher rates in *Elmo2* mutant mice? Could an earlier analysis of proliferation for example at E11.5 be worth investigating? Does the single-cell RNA seq. data provide any further insight into proliferation of EC and vSMCs?

Thank you very much for this comment. We totally agree with the reviewer's statement that proliferation of ECs and VSMCs needs to be increased in *Elmo2* mutants to accommodate the massive enlargement of the 3rd PAAs and carotid arteries. This is indeed what is shown by the quantification of the absolute number of EdU⁺ ECs and VSMCs at E12.5 (Supplementary Fig. 4b). The normalization to vessel perimeter indicates that this increased proliferation most likely reflects the change in vessel diameter and is unlikely to be driver of this enlargement (Supplementary Fig. 4c). This is consistent with previous studies showing that alterations in mechanical stretch¹⁸ and blood flow-derived forces¹⁹ can promote EC proliferation and remodeling.

As requested by the reviewer, we have now analyzed EC and VSMC proliferation earlier in development (Supplementary Fig. 4d-f). These results show that that the absolute and normalized number of proliferating vascular cells are not significantly different in *Elmo2* knockouts and littermate control embryos at E11.5, which also argues that the changes in proliferation at E12.5 are most likely a secondary response to 3rd PAA dilation.

The analysis of cell cycle regulators in our scRNA-seq data from E12.5 control and *Elmo2*^{-/-} embryos correlates well with the quantitative data obtained from the EdU labeling analysis, as more mutant ECs are found in S-phase (Revision Fig. 5a), whereas the proportion of the major cell populations is unchanged in control and *Elmo2*^{-/-} samples (Revision Fig. 5b).

Revision Fig. 5. Proportion of cells in S-phase and with respect to the total population analyzed by scRNA-seq

- Bar plot from scRNA-seq cell counts showing the proportion of proliferating cells from the endothelial (EC), mesenchymal stromal (MSC) and mural cell clusters in control and *Elmo2*^{-/-} E12.5 embryos.
- Bar plot from scRNA-seq cell counts showing the proportion that ECs, MSCs and mural cells represent with respect to the total number of cells analyzed in control and *Elmo2*^{-/-} samples.

It should be also noted that EC-specific inactivation of *Elmo2* is not able to induce any of the changes detected in global or neural crest-specific *Elmo2* mutants. Finally, *Wnt1-Cre2*-mediated gene inactivation spares the 3rd PAA and carotid artery endothelium, further arguing that changes in EC proliferation are secondary to the vascular dilation.

Minor points:

1. In order to better understand from which regions of the embryos the tissue sections have been taken, a schematic drawing or alternatively a line in the embryo whole mount picture would be helpful in Figure 1.

Thank you so much for this excellent suggestion. We have now incorporated a scheme reflecting the orientation of the plane and the region imaged for each developmental stage (Fig. 1f).

2. A schematic representation (or table) summarizing which cell types are targeted by the different Cre lines and a summary of phenotypes (including different time points if applicable) would further improve the paper.

In this revised version of the manuscript, Supplementary Table 1 (Summary of *in vivo* phenotypes upon *Elmo2* deletion) has been added, providing an overview of the phenotypic outcomes after global or cell-type specific genetic inactivation.

3. In Fig 1G, the Prox-1 positive jugular lymph sac seems enlarged and hyperplastic in *Elmo2* mutants. Are these vessels entirely normal?

Indeed, as addressed in our reply to Reviewer # 2 (major point 2), in some instances dilation of the jugular lymph sac or presence of red blood cells was observed. We interpret these findings as secondary effects caused by extravasation, edema and drastic changes in blood flow caused by stasis in the aneurysm lesions. In addition, the orientation of the samples during sectioning may also induce small variations in the observed size of vascular structures.

4. Along those lines, several other vessels (not only the carotid artery) appear enlarged in Supplementary Figure 6b.

Many thanks for pointing this out. In this case, the abnormal structure of the vessels is in part due to histological artifacts, as these images were obtained from cryosections. We have generated new data using vibratome sections with improved preservation of tissue integrity. The revised version of Supplementary Fig. 6 clearly reflects that the dilation phenotype affects primarily the carotid arteries. Data for other arteries is provided in Supplementary Fig. 2.

5. It would be appreciated if the time-lapse movies were merged and the speed was reduced to better follow the differences between control and *Elmo2* knockdown cells. The movies also appear to suffer from focus drift, which makes it difficult to appreciate the phenotypical differences between control and *Elmo2* knockdown cells.

This is also a fair point, which we have addressed. The experiment has been repeated using a different imaging setup in order to increase our capacity of accurately imaging cells even when their focal plane change during attachment. We have also increased the temporal resolution by acquiring more frames per time unit. In the new data set, the individual videos for the control and knockdown cells have been merged to ease the comparison.

6. In Supplementary Figure 1f the labeling of the carotid arteries is missing. It would be helpful to the reader if the labeling could be added, for consistency.

Thank you for alerting us to this omission. The missing labels are now included.

7. In Line 144/Figure 2e and f: Ectopic expression of SM22a in ECs look not fully convincing to me. It rather looks like that vascular smooth muscle cells invade the endothelial layer rather than ECs start to express vSMC specific genes. Maybe a zoom in panel and another color

combination e.g. green/violet could help to clarify this. Do these cells also express other vSMC markers?

The ectopic expression of VSMC-markers in the ECs lining the 3rd PAA or carotid arteries of *Elmo2*^{-/-} embryos is a phenomenon we have consistently observed and characterized by different means. We agree that the previous image may have not been the best possible representation though. We have now replaced the data using SOX17 for identification of arterial ECs instead of the pan-endothelial marker CD31 (Fig. 2d). The nuclear localization of this marker facilitates unambiguous observation of SM22 α expression in the endothelium of the mutant 3rd PAA. In addition, we present additional immunostaining data showing ectopic expression of α SMA and SM22 α in mutant ECs, consistent with our original description (Revision Fig. 6a and b).

Revision Fig. 6. Ectopic expression of VSMC markers in arterial ECs of *Elmo2*^{-/-} embryos

- Confocal image of 3rd pharyngeal arch artery (PAA) from control and *Elmo2*^{-/-} E12.5 embryos stained for ECs (CD31, green) and VSMCs (SM22 α , magenta). White arrowheads indicate ECs with ectopic expression of SM22 α . The images on the right are a high magnification view of the dashed-line boxes on the left panel. Scale bars, 25 μ m and 10 μ m (higher magnification).
- Confocal image 3rd PAA from control and *Elmo2*^{-/-} E12.5 embryos stained for ECs (CD31, grey) and VSMCs (α SMA, red). White arrowheads indicate ECs with ectopic expression of α SMA. Scale bars, 10 μ m.

Regarding the expression of other VSMC markers, our scRNA-seq data (Fig. 3g) highlights ectopic expression of *Acta2*, *Myl9* and *Ccn2* in *Elmo2*^{-/-} arterial ECs.

8. TH-IRES-Cre mediated *Elmo2* deletion is described but not shown (only the lineage tracing is shown. Supplementary Figure 10d). Could the authors add this data?

We now provide the detailed characterization of the recombination pattern and the absence of phenotypic changes in carotid artery diameter upon *TH-IRES-Cre*-mediated *Elmo2* deletion in Supplementary Fig. 11.

9. The authors use phospho-myosin light chain 2 (phMLC2) as a functional marker for vSMC contraction. Could the authors please add an appropriate reference for that? Based on

available single-cell datasets expression of Myl2 in vSMCs is rather weak. Maybe pMYL9 would be a better marker?

We appreciate this observation and agree with the reviewer that the available expression data indicates that *My19* may be a more suitable marker to analyze. We have therefore performed additional immunostaining experiments using an antibody against phospho-Myosin light chain 9 (phMLC9) both *in vivo* (Supplementary Fig. 3h) and *in vitro* (Supplementary Fig. 12d). In both cases the staining distribution and intensity in control and *Elmo2*-deficient samples is indistinguishable so that the original conclusions remain unaffected.

Reviewer #4 (Remarks to the Author):

Many thanks as well to this reviewer for contributing to the constructive suggestions.

REFERENCES

- 1 Gunst, S. J. & Zhang, W. Actin cytoskeletal dynamics in smooth muscle: a new paradigm for the regulation of smooth muscle contraction. *Am J Physiol Cell Physiol* **295**, C576-587 (2008). <https://doi.org:10.1152/ajpcell.00253.2008>
- 2 Tang, D. D. The Dynamic Actin Cytoskeleton in Smooth Muscle. *Adv Pharmacol* **81**, 1-38 (2018). <https://doi.org:10.1016/bs.apha.2017.06.001>
- 3 Yamin, R. & Morgan, K. G. Deciphering actin cytoskeletal function in the contractile vascular smooth muscle cell. *J Physiol* **590**, 4145-4154 (2012). <https://doi.org:10.1113/jphysiol.2012.232306>
- 4 Cipolla, M. J., Gokina, N. I. & Osol, G. Pressure-induced actin polymerization in vascular smooth muscle as a mechanism underlying myogenic behavior. *FASEB J* **16**, 72-76 (2002). <https://doi.org:10.1096/cj.01-0104hyp>
- 5 Rahman, A. *et al.* The small GTPase Rac1 is required for smooth muscle contraction. *J Physiol* **592**, 915-926 (2014). <https://doi.org:10.1113/jphysiol.2013.262998>
- 6 Alford, P. W., Nesmith, A. P., Seywerd, J. N., Grosberg, A. & Parker, K. K. Vascular smooth muscle contractility depends on cell shape. *Integr Biol (Camb)* **3**, 1063-1070 (2011). <https://doi.org:10.1039/c1ib00061f>
- 7 Bogunovic, N. *et al.* Impaired smooth muscle cell contractility as a novel concept of abdominal aortic aneurysm pathophysiology. *Sci Rep* **9**, 6837 (2019). <https://doi.org:10.1038/s41598-019-43322-3>
- 8 Garcia-Gonzalez, I. *et al.* iSuRe-HadCre is an essential tool for effective conditional genetics. *Nucleic Acids Res* **52**, e56 (2024). <https://doi.org:10.1093/nar/gkae472>
- 9 Loonstra, A. *et al.* Growth inhibition and DNA damage induced by Cre recombinase in mammalian cells. *Proc Natl Acad Sci U S A* **98**, 9209-9214 (2001). <https://doi.org:10.1073/pnas.161269798>
- 10 Wagenseil, J. E. & Mecham, R. P. Vascular extracellular matrix and arterial mechanics. *Physiol Rev* **89**, 957-989 (2009). <https://doi.org:10.1152/physrev.00041.2008>
- 11 Michel, J. B., Jondeau, G. & Milewicz, D. M. From genetics to response to injury: vascular smooth muscle cells in aneurysms and dissections of the ascending aorta. *Cardiovasc Res* **114**, 578-589 (2018). <https://doi.org:10.1093/cvr/cvy006>
- 12 Katsuda, S. & Kaji, T. Atherosclerosis and extracellular matrix. *J Atheroscler Thromb* **10**, 267-274 (2003). <https://doi.org:10.5551/jat.10.267>
- 13 McLean, S. E., Mecham, B. H., Kelleher, C. M., Mariani, T. J. & Mecham, R. P. Extracellular matrix gene expression in the developing mouse aorta. *Advances in Developmental Biology* **15**, 81-128 (2005). [https://doi.org:https://doi.org/10.1016/S1574-3349\(05\)15003-0](https://doi.org:https://doi.org/10.1016/S1574-3349(05)15003-0)

- 14 Beamish, J. A., He, P., Kottke-Marchant, K. & Marchant, R. E. Molecular regulation of contractile smooth muscle cell phenotype: implications for vascular tissue engineering. *Tissue Eng Part B Rev* **16**, 467-491 (2010). <https://doi.org/10.1089/ten.TEB.2009.0630>
- 15 Standley, P. R., Cammarata, A., Nolan, B. P., Purgason, C. T. & Stanley, M. A. Cyclic stretch induces vascular smooth muscle cell alignment via NO signaling. *Am J Physiol Heart Circ Physiol* **283**, H1907-1914 (2002). <https://doi.org/10.1152/ajpheart.01043.2001>
- 16 Zhu, J. H. *et al.* Cyclic stretch stimulates vascular smooth muscle cell alignment by redox-dependent activation of Notch3. *Am J Physiol Heart Circ Physiol* **300**, H1770-1780 (2011). <https://doi.org/10.1152/ajpheart.00535.2010>
- 17 Thomas, P. S., Kim, J., Nunez, S., Glogauer, M. & Kaartinen, V. Neural crest cell-specific deletion of Rac1 results in defective cell-matrix interactions and severe craniofacial and cardiovascular malformations. *Dev Biol* **340**, 613-625 (2010). <https://doi.org/10.1016/j.ydbio.2010.02.021>
- 18 Jufri, N. F., Mohamedali, A., Avolio, A. & Baker, M. S. Mechanical stretch: physiological and pathological implications for human vascular endothelial cells. *Vasc Cell* **7**, 8 (2015). <https://doi.org/10.1186/s13221-015-0033-z>
- 19 Campinho, P., Vilfan, A. & Vermot, J. Blood Flow Forces in Shaping the Vascular System: A Focus on Endothelial Cell Behavior. *Front Physiol* **11**, 552 (2020). <https://doi.org/10.3389/fphys.2020.00552>